# Federated Learning from Vision-Language Foundation Models: Theoretical Analysis and Method

**Bikang Pan**
ShanghaiTech University
panbk2023@shanghaitech.edu.cn

**Wei Huang** *
RIKEN Center for Advanced Intelligence Project
wei.huang.vr@riken.jp

**Ye Shi** *
ShanghaiTech University
shiye@shanghaitech.edu.cn

## Abstract

Integrating pretrained vision-language foundation models like CLIP into federated learning has attracted significant attention for enhancing generalization across diverse tasks. Typically, federated learning of vision-language models employs prompt learning to reduce communication and computational costs, i.e., prompt-based federated learning. However, there is limited theoretical analysis to understand the performance of prompt-based federated learning. In this work, we construct a theoretical analysis framework for prompt-based federated learning via feature learning theory. Specifically, we monitor the evolution of signal learning and noise memorization in prompt-based federated learning, demonstrating that performance can be assessed by the ratio of task-relevant to task-irrelevant coefficients. Furthermore, we draw an analogy between income and risk in portfolio optimization and the task-relevant and task-irrelevant terms in feature learning. Leveraging inspiration from portfolio optimization that combining two independent assets will maintain the income while reducing the risk, we introduce two prompts: global prompt and local prompt to construct a prompt portfolio to balance the generalization and personalization. Consequently, we showed the performance advantage of the prompt portfolio and derived the optimal mixing coefficient. These theoretical claims have been further supported by empirical experiments. Our code is available at: https://github.com/PanBikang/PromptFolio.git.

## 1 Introduction

Federated learning (FL) [33] stands out as a powerful framework that enables machine learning over decentralized data while maintaining data privacy and reducing reliance on centralized data repositories. Despite its advantages, the intensive computational and communication demands during the training phase often constrain the scalability of the models utilized. A transformative advancement in this field is the integration of prompt-based learning within the federated framework [42, 17, 13].

Prompt learning adapts models such as Contrastive Language-Image Pretraining (CLIP) [38] through minimal modifications, typically in the form of prompts or cues, guiding the model's predictions. As an emerging idea in machine learning, it has shown significant promise in various applications by allowing models to perform specialized tasks without undergoing complete retraining. Prompts effectively adjust pre-trained models for new tasks or datasets, which is particularly valuable in federated environments where data privacy and bandwidth constraints often limit conventional

---

*Corresponding author.

38th Conference on Neural Information Processing Systems (NeurIPS 2024).

training methods. A prominent example is CoOp [42], with its streamlined federated version known as PromptFL [17].

Despite the empirical success of prompt-based federated learning [28, 17, 16], theoretical analysis in this area remains limited. In this paper, we use feature learning theory [1] to propose an analytical framework for prompt-based federated learning with vision-language foundation models. Feature learning theory divides data into task-relevant and task-irrelevant features, allowing learnable weights to be expressed as a linear combination of these features. By introducing feature alignment across bimodal pretrained models, we demonstrate the relationship between learnable prompts and the features in latent space. To understand the process of signal learning and noise memorization, we use a two-stage analysis to examine the dynamics of coefficients for task-relevant and task-irrelevant features. Leveraging these coefficients, we demonstrate that prompt learning can be evaluated by comparing the ratio of task-relevant to task-irrelevant coefficients. In this way, we establish a theoretical foundation for prompt-based federated learning.

Additionally, we treat task-relevant coefficients and task-irrelevant coefficients as income and risk in investment portfolios [31, 32]. Inspired by investment portfolios, where combining two independent assets can reduce risk, we introduce two learnable prompts: a global prompt and a local prompt, and simply mix them to form a prompt portfolio to balance generalization and personalization. By mixing them to create a prompt portfolio, we balance the generalization and personalization under severe data heterogeneity. Leveraging the analysis framework we proposed, we prove the performance advantage of this mixed algorithm and derive the optimal coefficient. Besides, we use comprehensive experiments to support our theoretical results.

In this paper, our primary contributions are threefold:

- We present an analytical framework for prompt-based federated learning using vision-language foundation models. This framework aligns text and image features into a shared latent feature space and utilizes a two-stage analysis to understand the dynamics of prompt learning. By tracking the progression of signal learning and noise memorization, we show that the effectiveness of prompt-based federated learning can be measured by the ratio of task-relevant to task-irrelevant coefficients.

- Additionally, we introduce a prompt portfolio mechanism to address severe data heterogeneity and balance generalization and personalization. Within our proposed analytical framework, we draw an intuitive analogy between task-relevant features and income in portfolio optimization, as well as task-irrelevant features and risk. Consequently, we demonstrate that the combination of prompts performs better than using a single prompt, and we provide the optimal mixing ratio.

- The theoretical result has been empirically validated through rigorous experiments. Our results not only align with theoretical predictions but also consistently demonstrate the practical superiority of our approach in real-world scenarios.

## 2 Related Work

In this section, we examine prior research that serves as the basis for our study. Our investigation is divided into two primary areas: prompt-based federated learning and feature learning theory. Together, these fields create a complete context for our contributions.

### 2.1 Prompt-based federated learning

Prompt learning, initially developed in the field of natural language processing, has expanded its reach to vision language models. Examples include the CLIP model [38], which originally utilized manually engineered templates. However, more recent advancements have shifted towards developing prompts within a continuous embedding space. Innovations like CoOp [42] have refined the CLIP model by introducing continuous prompt vectors, sparking a surge of studies aimed at enhancing the efficiency of prompt learning and providing a solid foundation for further investigation [12].

To enhance the integration of global data and address challenges in scenarios with limited user data, FedPrompt [41] and PromptFL [17] have effectively integrated the concept of prompt learning with federated learning [4, 5, 27, 37, 23]. To tackle the statistical heterogeneity often found in client data,

pFedPrompt [16] introduces a non-parametric approach, providing each client with a personalized attention module. This module is designed to refine spatial visual features to better align with the unique characteristics of local data. Concurrently, pFedPG [40] introduces a novel prompt generator located at the server, which customizes prompts for each client, thus enhancing the personalization of the federated learning process. Additionally, a recent study, FedOTP [28], leverages optimal transport theory to improve the balance between achieving global consensus and allowing for local customization through a strategic combination of global and local prompts.

However, there is limited theoretical analysis of federated prompt learning. For instance, the theoretical examination of the CLIP model by [8] enhances prompt learning theory by exploring how CLIP learns transferable representations across various modalities and improves zero-shot transfer performance with a recently developed regularization technique. However, to the best of our knowledge, no research analyzes prompt learning within a federated setting that elucidates the cooperation between prompts. In this paper, we analyze how different prompts interact with feature learning theory and demonstrate the provable benefits of cooperation between global and local prompts.

## 2.2 Feature learning theory

Feature learning theory [1] has revolutionized our understanding of how machine learning models learn and represent information. Unlike other theories, feature learning accommodates substantial weight updates during gradient descent, enabling the network to uncover complex patterns in data. Feature learning theory has been successfully applied to various neural network architectures, including convolutional neural networks [6, 24], graph neural networks [18], and vision transformers [22, 26]. Moreover, feature learning theory has been used to analyze different training algorithms, such as Adam [43], momentum [26], out-of-distribution learning [7], and Mixup [44, 10]. Notably, [19] have analyzed the convergence and generalization in general federated learning. Despite progress in feature learning theory, the study of federated prompt learning is sparse. Our work uniquely addresses this gap by analyzing feature learning under prompt-based federated learning, providing crucial insights for their effective adaptation and optimization in such contexts.

## 3 Preliminary

**Notation** In our notation, vectors are represented by lowercase bold letters, matrices by uppercase bold letters, and scalars by regular, non-bold letters. The $\ell_2$-norm of a vector $\mathbf{v}$ is indicated as $|\mathbf{v}|_2$. The spectral norm of a matrix $\mathbf{A}$ is denoted by $|\mathbf{A}|_2$, and its Frobenius norm by $|\mathbf{A}|_F$. To compare the growth or decline of two sequences, we use standard asymptotic symbols like $O(\cdot)$, $o(\cdot)$, $\Omega(\cdot)$, and $\Theta(\cdot)$, which describe their behavior as they approach infinity. We introduce notations $\tilde{O}(\cdot)$, $\tilde{\Omega}(\cdot)$, and $\tilde{\Theta}(\cdot)$ to obscure logarithmic factors within these expressions. Notably, $\mathbb{1}(\cdot)$ represents the indicator function. Lastly, we represent sequences of integers as $[n] = \{1, 2, \ldots, n\}$ and sequences of elements such as vectors can also be represented similarly $\mathbf{v}_{[n]} = \{\mathbf{v}_1, \mathbf{v}_2, \ldots, \mathbf{v}_n\}$.

## 3.1 Prompt-based federated learning

In this part, we demonstrate how to fine-tune a learnable text prompt under a vision language pre-trained model. Here, we consider the classification task, where we assume that we have an image $\mathbf{x}$. The objective is to correctly classify the image into class $y \in [C]$, where the total number of classes is $C$. From the vision language pre-trained model, we expect that the latent spaces of the text encoder and image encoder are aligned. Thus, when we input different prompts, the text feature generated by the correct prompt will have the highest similarity with the image feature. We input a learnable prompt $\mathbf{p}$ and a fixed class prompt $\mathbf{p}_c \in \{\mathbf{p}_1, \cdots, \mathbf{p}_C\}$, which correspond to the classes, into the text encoder $h$. This process generates the text feature for class $c$: $\mathbf{h}_c = h(\mathbf{p}, \mathbf{p}_c)$. On the other hand, the image feature $\mathbf{g}$ is generated by the image encoder $g$: $\mathbf{g}_{k,i} = g(\mathbf{x}_{k,i}) \in \mathbb{R}^m$. We define the similarity function between the image feature $\mathbf{g}$ and the text feature $\mathbf{h}$ as $\boldsymbol{\rho} := [\rho_c] = \text{sim}(\mathbf{g}, \mathbf{h}_c)$. The training process mirrors traditional classification tasks, where the objective loss $\ell(\boldsymbol{\rho}, \mathbf{e}_y)$ is the distance between the similarity vector and the actual label $y$. Here, $\ell$ represents the loss function that measures the distance between two vectors, and $\mathbf{e}_y$ is the one-hot vector derived from the ground truth label $y$.

To illustrate the prompt-based federated learning framework, consider a federated system with a central server and $K$ clients. Assume client $k$ has $n_k$ training samples: $\{\mathbf{x}_{k,i}, y_{k,i}\}_{i=1}^{n_k}$. The learnable prompt in client $k$ is denoted as $\mathbf{p}_k$ and the learnable prompt is aggregated at each communication round.

# 4 Analysis Framework for Prompt-based Federated Learning

In this subsection, we present the analysis framework for prompt-based federated learning from vision-language pre-trained models. The central concept of this framework is the alignment of features between text and image encoders in the vision-language pre-trained model. To achieve this, we link the text encoder and image encoder through a shared latent feature space, described as follows.

**Feature representation and client distribution** Inspired by [6, 24, 19], we expect that the latent feature space will contain task-relevant and task-irrelevant features. In federated learning settings, the task-relevant features can be categorized into global task-relevant features $\boldsymbol{\mu}_G$ and local task-relevant features $\boldsymbol{\nu}_1, \cdots, \boldsymbol{\nu}_S$, where $S$ is the length of local task-relevant features. Additionally, the task-irrelevant features can be listed as $\boldsymbol{\xi}_1, \cdots, \boldsymbol{\xi}_L$, where $L$ is the length of task-irrelevant features. Here, we assume that the dimension of the latent space is $m$ and these features $\boldsymbol{\mu}_{(\cdot)}, \boldsymbol{\nu}_{(\cdot)}$, and $\boldsymbol{\xi}_{(\cdot)}$ are elements of $\mathbb{R}^m$. For simplicity, we assume that these features are orthogonal to each other. In our theoretical examination, we address a binary classification scenario with $y_{k,i} \in \{+1, -1\}$. We consider a scenario with $K$ clients, each linked to a distribution $\mathcal{D}_k, \forall k \in [K]$. Initially, we choose the signal vector $\boldsymbol{\mu}_k$ for client $k$ by sampling from $P(\boldsymbol{\nu}_1, \boldsymbol{\nu}_2, \ldots, \boldsymbol{\nu}_S)$, where $P$ represents a discrete distribution that assigns probabilities to each local task-relevant feature vector $\boldsymbol{\mu}_k := \boldsymbol{\nu}_s, s \in [S]$.

**Text encoder** Here, we suggest coupling the learnable prompt and the class prompt and propose the structure of the text encoder. By adopting a similar setting as [39], we suppose the generation of the text feature can be written as follows:

$$\mathbf{h}_{k,i} = h(\mathbf{p}_k, \mathbf{p}_{y_{k,i}}) = \sigma(\mathbf{W}\mathbf{p}_k + \mathbf{W}\mathbf{p}_{y_{k,i}}) - \sigma(-\mathbf{W}\mathbf{p}_k + \mathbf{W}\mathbf{p}_{y_{k,i}}), \tag{1}$$

where $\mathbf{W} \in \mathbb{R}^{m \times m_p}$ is the weight matrix, and $\mathbf{p}_{y_{k,i}} \in \mathbb{R}^{m_p}$ is the prompt linked to a ground truth class. In this definition, the introduction of $\mathbf{W}\mathbf{p}_{y_{k,i}}$ introduces nonlinearity between the trainable prompt and the class prompt while keeping the overall function nonlinear. Note that for a binary classification problem, the vector $\mathbf{p}_{y_{k,i}}$ takes $\mathbf{p}_1$ when $y_{k,i} = 1$ and $\mathbf{p}_{-1}$ when $y_{k,i} = -1$. To reveal the properties of the text encoder in the pre-trained model, we assume that the weight matrix $\mathbf{W}$ is composed of the following rows:

$$\mathbf{W} = [\boldsymbol{\mu}_G, \boldsymbol{\nu}_1, \cdots, \boldsymbol{\nu}_s, \cdots, \boldsymbol{\nu}_S, \boldsymbol{\xi}_1, \cdots, \boldsymbol{\xi}_L]^T. \tag{2}$$

The assumption of the weight matrix is inspired by [20], and the evidence of this assumption is listed in the Appendix D. In our analysis framework, we adapt FedAvg [33] as our prompt aggregation algorithm, which is named PrompFL [17]. The aggregation formula is given by:

$$\mathbf{p}_G^{(t+1)} \leftarrow \sum_{k=1}^{K} \frac{n_k}{n} \mathbf{p}_{G,k}^{(t)}, \tag{3}$$

where $n := \sum_k n_k$ denotes the total number of data samples across all clients.

**Image encoder** Let us consider the image network, represented as $\mathbf{g}_{k,i} = g(\mathbf{x}_{k,i}) \in \mathbb{R}^m$. We assume that the image encoder also aligns the feature space of the text encoder within the pre-trained model. As a result, we assume the image feature generated by data $\mathbf{x}_{k,i}$ in client $i$ can be expressed as follows:

$$\mathbf{g}_{k,i} = g(\mathbf{x}_{k,i}) = [y_{k,i}, \underbrace{0, \cdots}_{(s-1) \text{ zeros}}, y_{k,i}, \underbrace{\cdots, 0}_{(S-s) \text{ zeros}}, x_{k,i,1}, \cdots, x_{k,i,L}]^T \tag{4}$$

where $x_{k,i,l} \sim \mathcal{N}(0, \sigma_p^2), \forall l \in [L]$ represents the coefficient of task-irrelevant terms in the data, and $\sigma_p^2$ is the variance. This assumption implies that task-relevant features vary based on whether the label is positive or negative, whereas task-irrelevant features persist as arbitrary and unrelated to the label's polarity. The similarity score between an image $\mathbf{x}_{k,i}$ and class $y_{k,i}$ is expressed as

$\text{sim}(\mathbf{g}_{k,i}, \mathbf{h}_{k,i}) = \langle \mathbf{g}_{k,i}, \mathbf{h}_{k,i} \rangle$. Moreover, the objective of the training loss is designed to enhance the resemblance between the image feature $g(\mathbf{x}_{k,i})$ and the text feature created by the ground truth prompt $\mathbf{p}_{y_{k,i}}$.

**Signal-noise decomposition** Based on the above model, we introduce the signal-noise decomposition of the learnable prompt here. Note that the proofs of the following lemmas and theorems are listed in the appendix.

**Lemma 4.1 (Feature Representation).** *At the $t$-th iteration, the learnable prompt $\mathbf{p}_k^{(t)}$ for client $k$ and the aggregated prompt $\overline{\mathbf{p}}^{(t)}$ can be rewritten as a linear combination of features and prompt initialization:*

$$\mathbf{p}_k^{(t)} = \beta_k^{(t)} ||\boldsymbol{\mu}_G||_2^{-2} \boldsymbol{\mu}_G + \sum_{k'=1}^{K} (\alpha_{k,k'}^{(t)} \mathbf{p}_{k'}^{(0)} + \gamma_{k,k'}^{(t)} ||\boldsymbol{\mu}_{k'}||_2^{-2} \boldsymbol{\mu}_{k'}) + \sum_{l=1}^{L} \phi_{k,l}^{(t)} ||\boldsymbol{\xi}_l||_2^{-2} \boldsymbol{\xi}_l,$$

$$\overline{\mathbf{p}}^{(t)} = \overline{\beta}^{(t)} ||\boldsymbol{\mu}_G||_2^{-2} \boldsymbol{\mu}_G + \sum_{k=1}^{K} (\overline{\alpha}_k^{(t)} \mathbf{p}_k^{(0)} + \overline{\gamma}_k^{(t)} ||\boldsymbol{\mu}_k||_2^{-2} \boldsymbol{\mu}_k) + \sum_{l=1}^{L} \overline{\phi}_l^{(t)} ||\boldsymbol{\xi}_l||_2^{-2} \boldsymbol{\xi}_l. \tag{5}$$

*where $\alpha_{\cdot,\cdot}^{(t)}$ are the coefficients of initialization, $\beta_{\cdot}^{(t)}$ is the coefficient of global task-relevant features, $\gamma_{\cdot,\cdot}^{(t)}$ is the coefficient of local task-relevant features, $\phi_{\cdot,\cdot}^{(t)}$ are the coefficients of task-irrelevant features, and the overlined coefficients are the averaged versions of the original coefficients.*

Here, since the learnable prompts can be written as a linear combination of the features, we can analyze the dynamics of these coefficients to understand the learning progress of the prompts. The normalized factor such as $||\boldsymbol{\mu}_G||_2^{-2}$ is used to make the coefficient similar to the inner product of the prompts and the features, $\beta_k^{(t)} \approx \langle \mathbf{p}_k^{(t)}, \boldsymbol{\mu}_G \rangle$.

**Coefficient dynamics** Inspired by previous studies [6, 24, 19], we employ a two-phase analysis to track the dynamics of coefficients in prompt-based federated learning from vision-language foundation models. By analyzing the dynamics of the coefficients, we can obtain the feature learning procedure during training. This two-stage analysis allows us to establish the order of coefficients and explore how they are affected by the mixing parameter $\theta$. For the theorem and proof of this analysis, please refer to Appendix F.

**Theorem 4.2 (Training Dynamics).** *There exists a total number of local updates $T_1 = R_1 E = O(\eta^{-1} K n \sigma_p^2 \sigma_L^2)$ such that*

$$\overline{\beta}^{(T_1)} = \Theta(\overline{n} K SNR_G^2), \quad \overline{\gamma}_k^{(T_1)} = \Theta(\overline{n} \chi_k SNR_k^2), \quad \overline{\phi}_l^{(T_1)} = O(1) \quad \forall k \in [K], l \in [L]. \tag{6}$$

*Here, $\overline{n} = \sum_k n_k / K$ is the average number of data in each client, and $SNR_G = ||\boldsymbol{\mu}_G||/(\sigma_p \sqrt{m})$, $SNR_k = ||\boldsymbol{\mu}_k||/(\sigma_p \sqrt{m})$ denote the signal-to-noise ratio between the task-relevant feature and task-irrelevant feature. We define $\chi_k = \sum_{k'=1}^{K} \langle \boldsymbol{\mu}_k, \boldsymbol{\mu}_{k'} \rangle / ||\boldsymbol{\mu}_k||_2^2$.*

**Test performance evaluation with coefficients** Here, we suppose the classification output of the $i$-th data in client $k$ is the class corresponding to the highest similarity between the text feature and image feature, denoted as $\hat{y}_{k,i}$. To assess the algorithm's performance, we evaluate the error rate in the test procedure as our test loss $L_{\mathcal{D}}$:

$$L_{\mathcal{D}}(\overline{\mathbf{p}}) = \frac{1}{n} \sum_{k=1}^{K} \sum_{i=1}^{n_k} \mathbb{1}(\hat{y}_{k,i} = y_{k,i}). \tag{7}$$

The following theorem demonstrates that the test loss can be considered as the probability that a Gaussian random variable falls below zero, with the mean and variance influenced by the task-relevant and task-irrelevant coefficients.

**Theorem 4.3 (Test Loss).** *The expectation of test loss $L_{\mathcal{D}}$ of an algorithm can be treated as the probability*

$$\mathbb{E}[L_{\mathcal{D}}] := P(z < 0), \qquad z \sim \mathcal{N}(\mu, \sigma^2), \tag{8}$$

*where $\mu$ and $\sigma$ are functions of task-relevant and task-irrelevant coefficients, as defined in Appendix E.*

Drawing from this theorem and the properties of Gaussian distributions, an algorithm's performance can be evaluated by the ratio $\mu/\sigma$. This ratio highlights the influence of task-relevant and task-irrelevant features on test loss. Specifically, a higher task-relevant coefficient coupled with a lower task-irrelevant coefficient typically leads to better performance.

**Connection with portfolio optimization** The Markowitz mean-variance model is a famous framework for assembling a portfolio of assets such that the expected return is maximized for a given level of risk [31, 32]. This model characterizes assets by their expected returns and risks. It claims that the return of the whole portfolio is a proportionally weighted combination of the assets' returns, and the risk of the whole portfolio is a function of the correlations of the component assets. According to the properties of task-relevant and task-irrelevant coefficients, the task-relevant coefficient can be directly added, and the task-irrelevant feature follows the additive property of Gaussian random variables. Thus, we connect the task-relevant coefficient to the return and the task-irrelevant feature to the risk. This connection provides insight that the combination of prompts, i.e., a prompt portfolio, will lead to a higher ratio of task-relevant features to task-irrelevant features.

# 5 PromptFolio: Global-Local Prompt Portfolio for Federated Learning

Building on the significant connection between the feature learning process and portfolio optimization, we treat the prompt trained by CoOp and the prompt trained by PromptFL as the two prompt assets and propose a simple yet powerful mixing algorithm, PromptFolio [2]. For simplicity, we refer to the prompt trained by CoOp as the local prompt $\mathbf{p}_L$ and the prompt trained by PromptFL as the global prompt $\mathbf{p}_G$.

---

**Algorithm 1** (PromptFolio) Global-Local Prompt Portfolio

1: Initialize $\mathbf{p}_G$ and $\mathbf{p}_{L,k}$ for all clients $k$
2: $t \leftarrow 0$             ▷ Initialization of the iteration counter
3: **while** not converged **do**
4:     **for** each client $k$ in parallel **do**
5:        Send $\mathbf{p}_G^{(t)}$ to client $k$, $\mathbf{p}_{G,k}^{(t)} \leftarrow \mathbf{p}_G^{(t)}$
6:        **for** each sample $(\mathbf{x}_{k,i}, y_{k,i})$ in client $k$'s data **do**
7:           Compute $\mathbf{g}_{k,i} \leftarrow g(\mathbf{x}_{k,i})$
8:           **for** $c \leftarrow 1$ to $C$ **do**
9:              Compute $\mathbf{h}_{k,i,c} \leftarrow (1-\theta) \cdot h(\mathbf{p}_{G,k}^{(t)}, \mathbf{p}_c) + \theta \cdot h(\mathbf{p}_{L,k}^{(t)}, \mathbf{p}_c)$
10:              Compute similarity $\rho_{k,i,c} \leftarrow \text{sim}(\mathbf{g}_{k,i}, \mathbf{h}_{k,i,c})$
11:           **end for**
12:           Update $\mathbf{p}_{G,k}, \mathbf{p}_{L,k}$ by minimizing train loss $\ell(\boldsymbol{\rho}_{k,i}, \mathbf{e}_{y_{k,i}})$
13:        **end for**
14:        Send $\mathbf{p}_{G,k}^{(t+1)}$ to server
15:     **end for**
16:     Update $\mathbf{p}_G^{(t+1)} \leftarrow \sum_{k=1}^{K} \frac{n_k}{n} \mathbf{p}_{G,k}^{(t+1)}$        ▷ FedAvg to aggregate global prompt
17:     $t \leftarrow t + 1$
18: **end while**
19: **return** $\mathbf{p}_G, \mathbf{p}_{L,k}$ for all $k$

---

## 5.1 PromptFolio Method

The local learning process generates the local feature by including a specific local prompt, whereas the global learning process adopts a similar strategy but also uses FedAvg to compile learnable prompts from various clients. We enhance cooperation between the local and global learning processes by merging both local and global features to create the final text feature. The text feature is produced as follows:

$$\mathbf{h}_{k,i,c} = (1-\theta) \cdot h(\mathbf{p}_G, \mathbf{p}_c) + \theta \cdot h(\mathbf{p}_{L,k}, \mathbf{p}_c), \tag{9}$$

---

[2]PromptFolio is pronounced as /prɛmptˈfoʊlioʊ/.

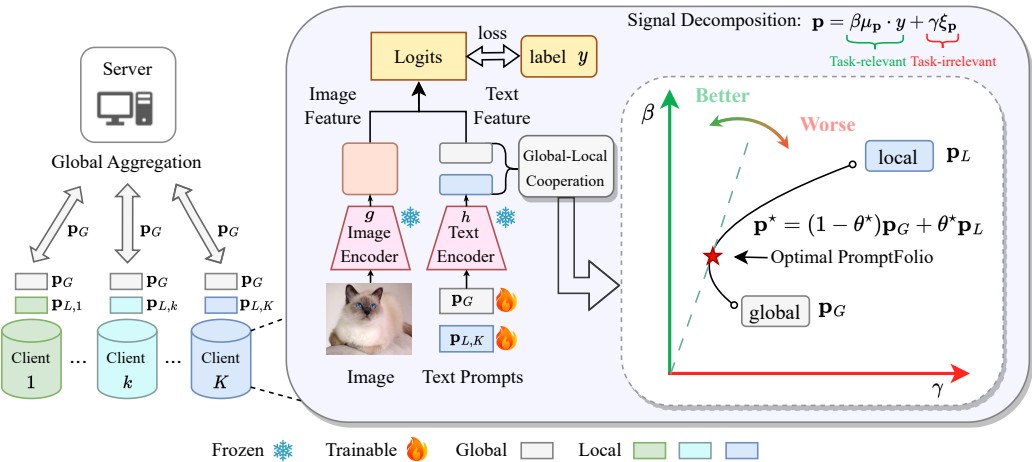

Figure 1: The image demonstrates the framework of the PromptFolio algorithm. The algorithm updates the global prompt and local prompt while keeping the weights of the fixed vision-language pretrained model unchanged. Additionally, it aggregates the global prompts from each client. The right side of the image intuitively demonstrates the advantages of global-local cooperation for performance when global and local are treated as two assets.

where $\theta \in [0, 1]$ serves as a coefficient to balance the mix of the two features, which addresses the balancing between personalization and generalization. The variation in the parameter $\theta$ influences the outcomes of the inference. Specifically, when $\theta = 0$, the algorithm reverts to PrompFL [17], whereas at $\theta = 1$, it shifts to CoOp [42]. Our approach consists of combining these features and using the resulting mixed feature to determine their similarity. This feature is subsequently utilized to evaluate the similarity between text and image features. Note that this algorithm differs from typical personalized algorithms [30] as it focuses on integrating text features instead of adjusting training weights. The framework of PromptFolio is described in Algorithm 1.

## 5.2 Analysis for PromptFolio

In this part, we offer a theoretical demonstration of the performance advantage of PromptFolio and the selection of the optimal mixing coefficient $\theta$. According to Theorem 4.3, each algorithm can be regarded as a Gaussian random variable. The test performance correlates with the ratio of task-relevant features to task-irrelevant features. This ratio enables us to analyze the test results of various learning algorithms.

Suppose that the coefficients of the prompt via local training at step $k$ are $\beta_k, \gamma_k$ and $\phi_k$, and the coefficients of the global prompt at step $k$ are $\overline{\beta}_k, \overline{\gamma}_k$ and $\overline{\phi}_k$. We define the mean and variance of the Gaussian variable corresponding to the local prompt as $\mu_k$ and $\sigma_k$, and the mean and variance of the Gaussian variable corresponding to the global prompt as $\overline{\mu}_k$ and $\overline{\sigma}_k$. Let $\rho = \Theta(1/k) \in [0, 1]$ be the correlation between the Gaussian variables of the local prompt and the global prompt. Here, we define $a := \frac{\mu_k}{\overline{\mu}_k} = \Theta\left(\frac{\beta_k + \gamma_k}{\overline{\beta}_k + \overline{\gamma}_k}\right) = \Theta\left(\frac{K\text{SNR}_G + K\text{SNR}_k}{K\text{SNR}_G + \chi_k \text{SNR}_k}\right)$ and $b := \frac{\sigma_k}{\overline{\sigma}_k} = O\left(\frac{\phi_k}{\overline{\phi}_k}\right) = O(K)$ as the ratio of different coefficients. Here, the order of $a$ and $b$ depends on the coefficient derived in Lemma F.3. As a result, we have the following theorem, and the proof can be referred to in Appendix E.3.

**Theorem 5.1 (PromptFolio Advantage).** *The mixed PromptFolio algorithm has a lower test loss than the mixing test loss of CoOp and PromptFL:*

$$L_{\mathcal{D}}((1-\theta)\mathbf{p}_G + \theta\mathbf{p}_L) \leq (1-\theta)L_{\mathcal{D}}(\mathbf{p}_G) + \theta L_{\mathcal{D}}(\mathbf{p}_L) \tag{10}$$

$$\forall \theta \in \left[0, \underset{[0,1]}{proj}\left(\frac{C_b - C_c}{2C_a}\right)\right], \quad where \begin{cases} C_a = (b-a)(b^2 + 2\rho b + 1) \\ C_b = (a+b)(b^2-1) - 4b(\rho b - 1) \\ C_c = (b-1)\sqrt{(a+b)^2(b+1)^2 - 8ab^2(\rho+1)^2} \end{cases}.$$

The results discussed demonstrate how combining global and local text features enhances performance and illustrate the optimal way to balance personalization with generalization. Here, $a$ and $b$ are the ratio of coefficients and reveal how the global feature and local feature interact. Drawing on principles from portfolio optimization, which involves blending two assets that are not perfectly correlated, we can construct a portfolio that maximizes returns while minimizing risk. Given the characteristics of Gaussian random variables, we intuitively correlate the coefficient of task-relevant features with returns and the coefficient of task-irrelevant features with risk. Thus, the first part of Theorem 5.1 provides a rationale that a well-balanced portfolio of prompt features can significantly improve performance.

Similar to the portfolio optimization problem, we can also derive the optimal mixing coefficient $\theta$.

**Theorem 5.2** (**Optimal Mixing Coefficient**). *The optimal mixing coefficient $\theta^\star$ follows*

$$\theta^\star = \underset{[0,1]}{proj} \left( \frac{a - \rho b}{(a + b^2) - \rho b(a + 1)} \right). \tag{11}$$

Theoretically, if we further simplify the mixing coefficient with the order of $a$, $b$ and $\rho$, then we get that

$$\theta^\star = \Theta \left( \frac{(K - \chi_k)\mathrm{SNR}_k}{(K^2 - 1)(K\mathrm{SNR}_G + \chi_k\mathrm{SNR}_k)} \right). \tag{12}$$

In this theorem, we note that a lower $\chi_k$ indicates greater data heterogeneity, which lead to a higher $\theta^\star$. This observation aligns with the intuition that, due to the non-i.i.d. distribution of data, the model should incorporate more local information, thereby making the optimal $\theta$ closer to 1.

# 6 Experiments

In this section, we conduct experiments with the CLIP model to empirically demonstrate the performance advantages of PromptFolio. Specifically, the image network $g$ and the text network $h$ are components of a pre-trained CLIP model. By evaluating results obtained using various mixing coefficients across different datasets, data distribution, and client number, we align theory with practice. We use the Dirichlet distribution to manage data heterogeneity and employ FedAvg as the aggregation strategy. The experiment is conducted on the CIFAR-100 dataset by default, with the model trained for 10 epochs locally and the results evaluated over 100 communication rounds.

## 6.1 Performance evaluation on various datasets

In this section, we observe that the combination of global and local algorithms outperforms both the prompt-based federated learning with FedAvg and individual prompt learning approaches. Under the CLIP model setting, the global and local prompt learning algorithm degenerates to PromptFL and CoOp, respectively. To explore why this global-local collaboration is more effective than either approach alone, we evaluate the accuracy of various mixing coefficients $\theta$ across different datasets. We use CIFAR-100 [25], DomainNet [36], Office-Caltech10 [15], OxfordPets [35], and DTD [11], adopting $\theta = 0.2$ as the general mixing coefficient for our algorithm. The quantitative results are shown in Table 1. From this table, it is evident that blending global and local prompts consistently leads to enhanced accuracy, with the accuracy curve also showing a peak. Further performance evaluations can be found in Appendix A.

Table 1: Accuracy of CoOp, PromptFL, PromptFolio on different datasets.

|  | **Cifar100** | **DomainNet** | **Office-Cal10** | **OxfordPets** | **DTD** |
|---|---|---|---|---|---|
| CoOp | $76.88 \pm 0.07$ | $91.83 \pm 0.13$ | $97.10 \pm 0.20$ | $87.85 \pm 0.32$ | $56.39 \pm 0.48$ |
| PromptFL | $78.16 \pm 0.16$ | $92.72 \pm 0.16$ | $95.51 \pm 2.62$ | $88.91 \pm 0.72$ | $70.99 \pm 0.32$ |
| PromptFolio | $\mathbf{80.17 \pm 0.05}$ | $\mathbf{93.04 \pm 0.09}$ | $\mathbf{97.24 \pm 0.11}$ | $\mathbf{92.17 \pm 0.32}$ | $\mathbf{71.32 \pm 0.49}$ |

## 6.2 Performance evaluation under various data heterogeneity

We then conduct the experiment over different data distributions. By varying the parameters of the Dirichlet distribution exponentially from 0.01 to 10, we controlled the heterogeneity of the data. A

larger $\alpha$ indicates that the data is closer to an i.i.d. distribution. Using 10 users, we performed our experiments, and the results are shown in Figure 2(a).

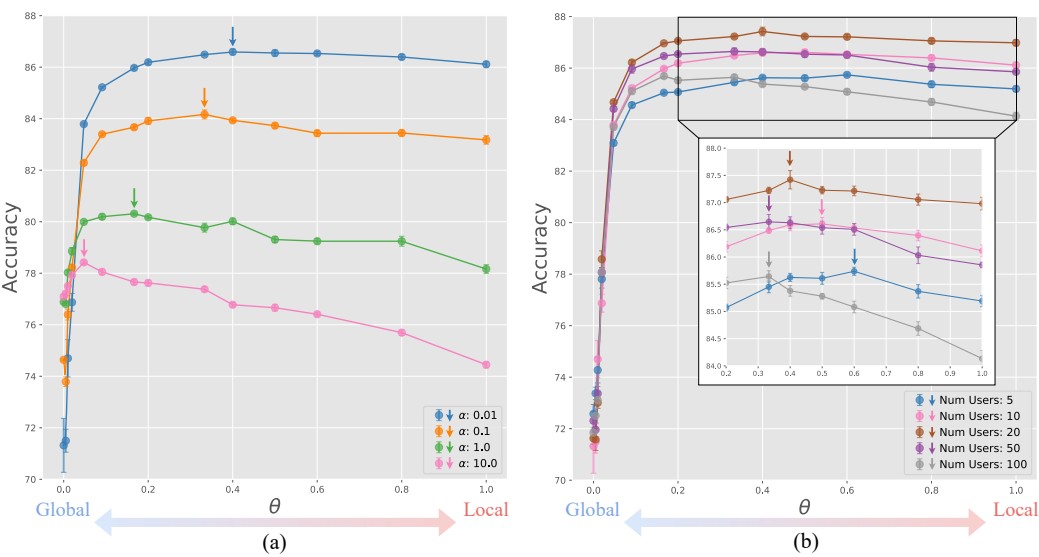

Figure 2: The x-axis represents the mixing coefficients, which range from 0 to 1, and the y-axis shows the accuracy of the test set after training. The left figure depicts the result under different data distributions, and the right figure reveals the result under different users.

From Figure 2(a), we observe that hybrid approaches outperform solitary methods, consistent with our theoretical analysis. Additionally, higher $\alpha$ values, indicating a more uniform distribution, generally result in a superior global model compared to local models tailored for specific users. This finding supports our conclusion that a more IID distribution leads to $\chi_k$ being higher and closer to $K$, resulting in a higher task-relevant to task-irrelevant coefficient ratio, and thus better performance. Conversely, in scenarios where data is highly non-IID, there is a preference for using more local models to maintain high accuracy, which aligns with our analysis of the optimal mixing coefficient $\theta^\star$.

## 6.3 Performance evaluation with different client number

Furthermore, we conducted experiments with different numbers of users on the CIFAR-100 dataset, keeping the Dirichlet distribution parameter fixed at $\alpha = 0.01$, which represents a pathological non-i.i.d. distribution. The number of users varies from 5 to 100, and the results are shown in Figure 2(b). From these results, we observe that the trend of the mixing strategy outperforming the independent global and local algorithms remains consistent, regardless of the number of users. As the number of users increases, the optimal $\theta$ shifts closer to zero, indicating that with more users, each client's information becomes less significant, necessitating more global information. Additionally, the accuracy first increases and then decreases with the number of users, suggesting that there is an optimal number of users, consistent with theoretical results.

## 7 Conclusion

This work presents a thorough theoretical and empirical exploration of prompt-based federated learning, integrating vision-language foundation models such as CLIP. By developing an analytical framework based on feature learning theory, we have examined the dynamics of signal learning and noise memorization specific to federated settings, providing a robust mechanism to evaluate the effectiveness of prompt-based learning strategies. Notably, our introduction of PromptFolio, which combines global and local prompts into a prompt portfolio, offers an approach to balancing generalization with personalization, drawing an innovative parallel with portfolio optimization in

finance. This approach balances generalization with personalization, supported by an optimal mixing coefficient from our theoretical framework to tailor adaptability in various federated settings. Empirical tests confirm our method's superiority, aligning with theoretical insights and outperforming traditional federated learning approaches. Limitations include a simplified text model with a single activation function, suggesting future work with more complex models to better capture deep network behaviors in federated environments.

## Acknowledgement

This work was supported by NSFC (No.62303319), Shanghai Sailing Program (22YF1428800), Shanghai Local College Capacity Building Program (23010503100), ShanghaiTech AI4S Initiative SHTAI4S202404, Shanghai Frontiers Science Center of Human-centered Artificial Intelligence (ShangHAI), MoE Key Laboratory of Intelligent Perception and Human-Machine Collaboration (ShanghaiTech University) and Shanghai Engineering Research Center of Intelligent Vision and Imaging. Wei Huang was supported by JSPS KAKENHI Grant Number 24K20848. Additionally, we would like to thank Leqi Zhou for her contributions to proofreading this paper.

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

**Supplementary organization:**

# A   Performance evaluation

In this paper, we adopt a similar setting to the experimental result in [28]. The experiments are conducted on a cluster with 2 Intel Xeon 5218R, 512GB memory, and 8 NVIDIA Tesla A40 GPUs 48GB. Here, we use Food101 [3], DTD [11], Caltech101 [14], Flowers102 [34], OxfordPets [35] to evaluate our algorithm. The baseline algorithms are CoOp [42], PromptFL [17], the variants of PromptFL [9, 29, 2, 21] and FedTPG [37]. We use Dirichlet distribution with parameter $\alpha = 0.3$ to split the dataset. Consider we use 10 users with 100 communication rounds to assess our algorithm. 10 epochs are conducted each communication round. There are 8 prompts that are randomly initialized during prompt learning. The accuracy is shown in Table 2.

|  | Food101 | DTD | Caltech101 | Flowers102 | OxfordPets |
|---|---|---|---|---|---|
| CoOp | 83.09±0.58 | 56.40±0.53 | 90.62±0.72 | 69.03±1.04 | 90.18±0.74 |
| PromptFL | 85.15±0.25 | 51.99±1.41 | 93.47±0.30 | 74.47±1.40 | 91.08±0.53 |
| PromptFL+FT | 80.85±0.27 | 50.72±1.17 | 89.04±0.59 | 69.05±1.94 | 87.12±1.15 |
| PromptFL+FedProx | 85.48±0.19 | 52.38±1.95 | 93.57±0.46 | 74.27±1.40 | 91.07±0.41 |
| PromptFL+FedPer | 82.20±1.03 | 58.14±0.37 | 91.70±0.58 | 71.71±0.53 | 90.65±0.69 |
| PromptFL+FedAMP | 83.24±0.52 | 56.40±0.53 | 91.38±0.21 | 70.80±0.42 | 90.56±0.43 |
| FedTPG | **86.59±0.03** | 52.01±0.80 | 93.38±0.12 | 72.91±1.23 | 90.98±0.24 |
| PromptFolio (Ours) | 86.50±1.34 | **61.04±0.69** | **93.59±0.39** | **74.61±0.53** | **92.08±0.14** |

Table 2: Comparison of different methods on various datasets.

# B   Further ablation study

This paper further conduct experiments on different shot numbers and different backbones. The results in shown as Figure 3.

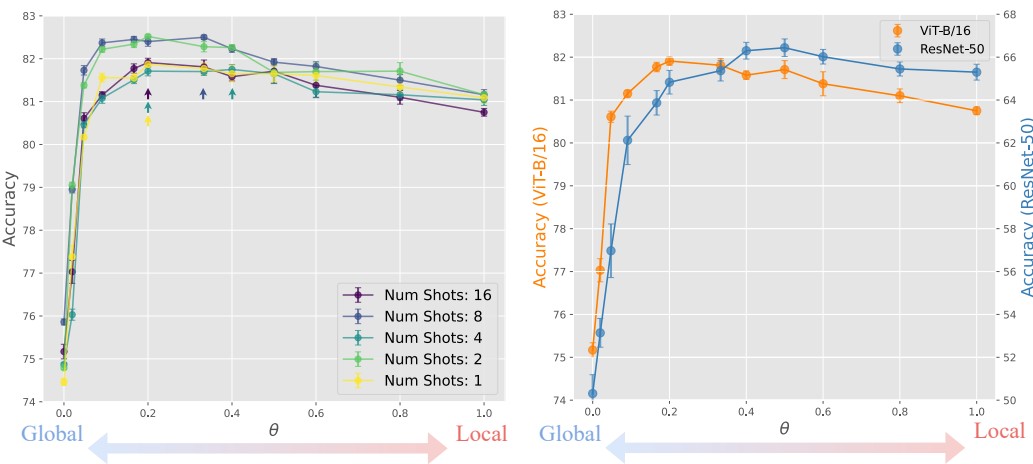

Figure 3: The accuracy curve among different shot numbers (left) and different backbones (right).

**Different shots** As shown in Figure 3(a), we tested the accuracy using different numbers of shots, ranging from 16-shot to 1-shot. It can be observed that the optimal coefficient remains stable and close to 0.2, demonstrating the robustness of our algorithm across various shot numbers..

**Different backbones** Figure 3(b) illustrates the accuracy using different vision backbones of CLIP, namely ViT-B/16 and ResNet-50. In this experiment, a similar trend is observed, where accuracy first increases and then decreases with the mixing coefficient, regardless of the backbone used. This outcome indicates that our theoretical framework is consistent across different backbones, further validating its applicability.

## C   Introduction to feature learning

Feature learning theory is a new theoretical framework proposed recently. The seminal work [1] proposed feature learning theory to understand ensemble, knowledge distillation, and self-distillation in deep learning. They show that when data has a structure containing features, the network's dynamics can be tracked during gradient training, which can further be used to characterize generalization. Later, this theory was further systematized and standardized by [7], forming a fundamental theoretical framework to explain benign overfitting. Since then, feature learning theory has been widely used to understand algorithms [8, 22] and techniques [20] in deep learning, forming a comprehensive theoretical system.

To provide more details on the general feature learning process:

1. **Defining the feature space**: As outlined in Section 4, features are categorized as task-relevant and task-irrelevant. For example, in an image of a cat, features representing the cat itself are task-relevant, while background features are task-irrelevant.

2. **Parameter representation**: As described in Lemma 4.1, learnable parameters can be decomposed into the feature space. In our framework, the learnable prompts are expressed as a linear combination of task-relevant and task-irrelevant features.

3. **Learning dynamics**: Theorem F.3 examines the learning dynamics of coefficients of feature learning, where coefficients are defined by the weight decomposition into the feature space, providing valuable insights into the learning process. When task-relevant features dominate, the network learns the target effectively and demonstrates good performance.

4. **Generalization bound**: By leveraging the learning dynamics of the coefficients, we can demonstrate the generalization bound or test performance post-training. In our work, we connect the performance of prompt fine-tuning with the ratio between task-relevant coefficients and task-irrelevant coefficients.

## D   Evidence of assumption

Here's the evidence supporting the second assumption that the features of the pretrained model are orthogonal. We focus on the final projection layer of the text encoder in CLIP [38] and calculate its cosine similarity between each row. The statistical distribution of the cosine similarity is shown in Figure D.

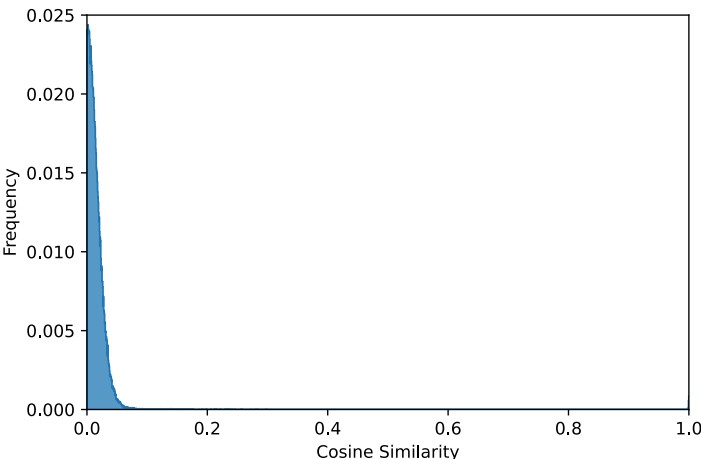

Figure 4: The distribution of cosine similarity between different rows of the final projection layer in CLIP [38].

Here, we find that nearly all of the cosine similarity are smaller than 0.1, which is close to orthogonal. This evidence support our theoretical assumption. Except the evidence of cosine similarity, the

previous research [20] also make assumption of orthonormal rows of weight matrix to the multi-modality latent space.

# E   Analysis framework

In this paper, our analysis will be made under the following assumptions:

**Assumption E.1.**  Suppose that:

1. The dimension of latent space $m$ is sufficiently large $m = \tilde{\Omega}(\overline{n})$.

2. The number of training samples and clients follows $\overline{n} = \Omega(\text{polylog(d)})$, $K = \Theta(1)$. The federated network shares the same initialization.

3. The learning rate $\eta \leq \tilde{O}(\frac{K}{E} \min\{||\boldsymbol{\mu}||_2^2, \sigma_p^{-2} m^{-1}\})$ and the standard deviation of network weight initialization $\sigma_0 \leq \tilde{O}(mn) \cdot \min\{(||\boldsymbol{\mu}||_2^2, \sigma_p \sqrt{d}^{-1})\}$

For the first assumption, we assume the dimension of latent space is large enough to make our analysis ensure an over-parameterized setting. The second assumption provides statistical properties to the training data and network weight initialization. The third assumption of a small learning rate and small initialization guarantees that gradient descent effectively minimizes the training loss.

Note that this analysis framework can be directly applied to our algorithm PromptFolio. To achieve a basic federated version of prompt learning, i.e., PromptFL, we apply the learnable prompt $\mathbf{p}$ as the global prompt $\mathbf{p}_G$ in PromptFolio and apply the mixing coefficient $\theta = 0$, we obtain the theoretical result of PromptFL. Similarly, we can apply the learnable prompt as the local prompt $\mathbf{p}_L$ and apply the mixing coefficient $\theta = 1$, we obtain the theoretical result of CoOp.

## E.1   Gradient update analysis

This section discusses the computational approach used to analyze the performance of the algorithm, focusing on the gradient calculations essential for optimizing the model parameters. To calculate the gradient, we first need to find the partial derivatives of $\mathbf{sim}(\mathbf{g}_{k,i}, \mathbf{h}_{k,i})$ with respect to $p_G$ and $p_L$. Let $\sigma'_{G,r,k,i}$ and $\sigma'_{L,r,k,i}$ are the $r$-th diagonal elements in $\sigma'(\mathbf{w}_r^T \mathbf{p}_{G,k} + \mathbf{w}_r^T \mathbf{p}_{y_{k,i}}) + \sigma'(-\mathbf{w}_r^T \mathbf{p}_{G,k} + \mathbf{w}_r^T \mathbf{p}_{y_{k,i}})$ and $\sigma'(\mathbf{w}_r^T \mathbf{p}_{L,k} + \mathbf{w}_r^T \mathbf{p}_{y_{k,i}}) + \sigma'(-\mathbf{w}_r^T \mathbf{p}_{L,k} + \mathbf{w}_r^T \mathbf{p}_{y_{k,i}})$ respectively:

$$\frac{\partial \mathbf{sim}(\mathbf{g}_{k,i}, \mathbf{h}_{k,i})}{\partial \mathbf{p}_{G,k}} = (1 - \theta)(\mathbf{W}^T(\sigma'(\mathbf{W}\mathbf{p}_{G,k} + \mathbf{W}\mathbf{p}_{y_{k,i}}) + \sigma'(-\mathbf{W}\mathbf{p}_{G,k} + \mathbf{W}\mathbf{p}_{y_{k,i}}))) \cdot g(\mathbf{x}_{k,i}),$$

$$\frac{\partial \mathbf{sim}(\mathbf{g}_{k,i}, \mathbf{h}_{k,i})}{\partial \mathbf{p}_{L,k}} = \theta(\mathbf{W}^T(\sigma'(\mathbf{W}\mathbf{p}_{L,k} + \mathbf{W}\mathbf{p}_{y_{k,i}}) + (\sigma'(-\mathbf{W}\mathbf{p}_{L,k} + \mathbf{W}\mathbf{p}_{y_{k,i}}))) \cdot g(\mathbf{x}_{k,i}). \quad (13)$$

Using the logistic loss $L^{\mathcal{T}}$ for client $k$, the goal is to minimize the subsequent loss function to achieve supervised fine-tuning of the text prompts:

$$L_k^{\mathcal{T}}(\mathbf{p}_{G,k}; \mathbf{p}_{L,k}) = -\frac{1}{n_k} \sum_{i=1}^{n_k} \log(1 + \exp(\mathbf{sim}(\mathbf{g}_{k,i}, \mathbf{h}_{k,i}))). \quad (14)$$

Now, we can use these partial derivatives to compute the gradients with respect to $\mathbf{p}_G$ and $\mathbf{p}_L$:

$$\nabla_{\mathbf{p}_{G,k}} L_k^{\mathcal{T}}(\mathbf{p}_{G,k}) = -\frac{1}{n_k} \sum_{i=1}^{n_k} \frac{\exp(\mathbf{sim}(\mathbf{g}_{k,i}, \mathbf{h}_{k,i}))}{1 + \exp(\mathbf{sim}(\mathbf{g}_{k,i}, \mathbf{h}_{k,i}))} \frac{\partial \mathbf{sim}(\mathbf{g}_{k,i}, \mathbf{h}_{k,i})}{\partial \mathbf{p}_{G,k}},$$

$$\nabla_{\mathbf{p}_{L,k}} L_k^{\mathcal{T}}(\mathbf{p}_{L,k}) = -\frac{1}{n_k} \sum_{i=1}^{n_k} \frac{\exp(\mathbf{sim}(\mathbf{g}_{k,i}, \mathbf{h}_{k,i}))}{1 + \exp(\mathbf{sim}(\mathbf{g}_{k,i}, \mathbf{h}_{k,i}))} \frac{\partial \mathbf{sim}(\mathbf{g}_{k,i}, \mathbf{h}_{k,i})}{\partial \mathbf{p}_{L,k}}. \quad (15)$$

To simplify the update process, we can rewrite it using summation notation. Let $\ell'_{k,i}$ denote $\frac{\exp(\mathbf{sim}(\mathbf{g}_{k,i},\mathbf{h}_{k,i}))}{1+\exp(\mathbf{sim}(\mathbf{g}_{k,i},\mathbf{h}_{k,i}))}$:

$$\nabla_{\mathbf{p}_{G,k}}L_k^{\mathcal{T}}(\mathbf{p}_{G,k}) = -\frac{1-\theta}{n_k}\sum_{i=1}^{n_k}\ell'_{k,i}(\mathbf{W}^T(\sigma'(\mathbf{W}\mathbf{p}_{G,k}+\mathbf{W}\mathbf{p}_{y_{k,i}})+\sigma'(-\mathbf{W}\mathbf{p}_{G,k}+\mathbf{W}\mathbf{p}_{y_{k,i}})))\cdot g(\mathbf{x}_{k,i})$$

$$= -\frac{1-\theta}{n_k}\sum_{i=1}^{n_k}\sum_{r=1}^{m}\ell'_{k,i}x_{r,k,i}\sigma'_{G,r,k,i}\mathbf{w}_r, \tag{16}$$

$$\nabla_{\mathbf{p}_{L,k}}L_k^{\mathcal{T}}(\mathbf{p}_{L,k}) = -\frac{\theta}{n_k}\sum_{i=1}^{n_k}\ell'_{k,i}(\mathbf{W}^T(\sigma'(\mathbf{W}\mathbf{p}_{L,k}+\mathbf{W}\mathbf{p}_{y_{k,i}})+\sigma'(-\mathbf{W}\mathbf{p}_{L,k}+\mathbf{W}\mathbf{p}_{y_{k,i}})))\cdot g(\mathbf{x}_{k,i})$$

$$= -\frac{\theta}{n_k}\sum_{i=1}^{n_k}\sum_{r=1}^{m}\ell'_{k,i}x_{r,k,i}\sigma'_{L,r,k,i}\mathbf{w}_r. \tag{17}$$

In the above equations, $x_{r,k,i}$ represents the $r$-th row of $g(\mathbf{x}_{k,i})$. Note that $\mathbf{w}_r^T$ represents the $r$-th row of the weight matrix $\mathbf{W}$.

## E.2  Signal-noise decomposition

In this subsection, we provide the proof of signal-noise decomposition as mentioned in Lemma 4.1. Note that the local update of global prompt coefficients can be generalized to local prompts by changing the subscript $G$ to $L$. The coefficient $\beta_{G/L,k}$ denotes the coefficient of the global prompt at client $k$. The coefficient $\gamma_{G/L,k,k'}$ denotes the coefficient of the local prompt $k'$ at client $k$. The coefficient $\phi_{G/L,k,k',i}$ denotes the coefficient of client $k'$'s $i$-th task-irrelevant feature at client $k$.

Here, we suppose that the weight of the pretrained model consists of two kinds of weights, task-relevant weight $\boldsymbol{\mu}$ and task-irrelevant weight $\boldsymbol{\xi}$. Without loss of generality, we can rewrite the gradient descent update of the prompt as follows:

$$\mathbf{p}_{G,k}^{(t)} = \beta_{G,k}^{(t)}\|\boldsymbol{\mu}_G\|_2^{-2}\boldsymbol{\mu}_G + \sum_{k'=1}^{K}(\alpha_{G,k,k'}^{(t)}\mathbf{p}_{G,k'}^{(0)}+\gamma_{G,k,k'}^{(t)}\|\boldsymbol{\mu}_{k'}\|_2^{-2}\boldsymbol{\mu}_{k'}) + \sum_{l=1}^{L}\phi_{G,k,l}^{(t)}\|\boldsymbol{\xi}_l\|_2^{-2}\boldsymbol{\xi}_l, \tag{18}$$

$$\overline{\mathbf{p}}_G^{(t)} = \overline{\beta}_G^{(t)}\|\boldsymbol{\mu}_G\|_2^{-2}\boldsymbol{\mu}_G + \sum_{k=1}^{K}(\overline{\alpha}_{G,k}^{(t)}\mathbf{p}_{G,k}^{(0)}+\overline{\gamma}_{G,k}^{(t)}\|\boldsymbol{\mu}_k\|_2^{-2}\boldsymbol{\mu}_k) + \sum_{l=1}^{L}\overline{\phi}_{G,l}^{(t)}\|\boldsymbol{\xi}_l\|_2^{-2}\boldsymbol{\xi}_l, \tag{19}$$

$$\mathbf{p}_{L,k}^{(t)} = \beta_{L,k}^{(t)}\|\boldsymbol{\mu}_G\|_2^{-2}\boldsymbol{\mu}_G + \alpha_{L,k,k}^{(t)}\mathbf{p}_{L,k}^{(0)}+\gamma_{L,k,k}^{(t)}\|\boldsymbol{\mu}_k\|_2^{-2}\boldsymbol{\mu}_k + \sum_{l=1}^{L}\phi_{L,k,l}^{(t)}\|\boldsymbol{\xi}_l\|_2^{-2}\boldsymbol{\xi}_l. \tag{20}$$

According to the update formula of the gradient descent, we have the update formula of the global prompt and the local prompt:

$$\mathbf{p}_{G,k}^{(t+1)} = \mathbf{p}_{G,k}^{(t)} - \eta\nabla_{\mathbf{p}_{G,k}}L^{\mathcal{T}}(\mathbf{p}_{G,k}^{(t)})$$

$$= \mathbf{p}_{G,k}^{(t)} + \frac{\eta}{n_k}(1-\theta)\sum_{i=1}^{n_k}\sum_{r=1}^{m}\ell'_{k,i}x_{r,k,i}\sigma'_{G,r,k,i}\mathbf{w}_r, \tag{21}$$

$$\mathbf{p}_{L,k}^{(t+1)} = \mathbf{p}_{L,k}^{(t)} - \eta\nabla_{\mathbf{p}_{L,k}}L^{\mathcal{T}}(\mathbf{p}_{L,k}^{(t)})$$

$$= \mathbf{p}_{L,k}^{(t)} + \frac{\eta}{n_k}\theta\sum_{i=1}^{n_k}\sum_{r=1}^{m}\ell'_{k,i}x_{r,k,i}\sigma'_{L,r,k,i}\mathbf{w}_r. \tag{22}$$

As a result, for the row corresponding to the global feature line, we have the update formula of the coefficients:

$$\beta_{G,k}^{(t+1)} = \beta_{G,k}^{(t)} + \frac{\eta}{n_k}(1-\theta)\cdot\sum_{i=1}^{n_k}\ell_{k,i}'^{(t)}\cdot\sigma_{G,r,k,i}'^{(t)}\cdot\|\boldsymbol{\mu}_G\|_2^2, \tag{23}$$

$$\beta_{L,k}^{(t+1)} = \beta_{G,k}^{(t)} + \frac{\eta}{n_k}\theta\cdot\sum_{i=1}^{n_k}\ell_{k,i}'^{(t)}\cdot\sigma_{G,r,k,i}'^{(t)}\cdot\|\boldsymbol{\mu}_G\|_2^2. \tag{24}$$

For the row corresponding to the local feature line, we have the update formula of the coefficients:

$$\gamma_{G,k,k'}^{(t+1)} = \gamma_{G,k,k'}^{(t)} + \frac{\eta}{n_{k'}}(1-\theta) \cdot \sum_{i=1}^{n_k} \ell_{k',i}'^{(t)} \cdot \sigma_{G,r,k',i}'^{(t)} \cdot ||\boldsymbol{\mu}_{k'}||_2^2 \mathbf{I}(k'=k), \tag{25}$$

$$\gamma_{G,k,k'}^{(t+1)} = \gamma_{G,k,k'}^{(t)} + \frac{\eta}{n_{k'}}\theta \cdot \sum_{i=1}^{n_k} \ell_{k',i}'^{(t)} \cdot \sigma_{G,r,k',i}'^{(t)} \cdot ||\boldsymbol{\mu}_{k'}||_2^2 \mathbf{I}(k'=k). \tag{26}$$

For the row corresponding to the task-irrelevant feature line, we have the update formula of the coefficients:

$$\phi_{G,k,l}^{(t+1)} = \phi_{G,k,l}^{(t)} + \frac{\eta}{n_k}(1-\theta) \cdot \sum_{i=1}^{n_k} \ell_{k,i}'^{(t)} \cdot \sigma_{G,r,k,i}'^{(t)} \cdot y_{k,i} \cdot x_{k,i,l} \cdot ||\boldsymbol{\xi}_l||_2^2, \tag{27}$$

$$\phi_{L,k,l}^{(t+1)} = \phi_{L,k,l}^{(t)} + \frac{\eta}{n_k}\theta \cdot \sum_{i=1}^{n_k} \ell_{k,i}'^{(t)} \cdot \sigma_{G,r,k,i}'^{(t)} \cdot y_{k,i} \cdot x_{k,i,l} \cdot ||\boldsymbol{\xi}_l||_2^2. \tag{28}$$

To analyze the increase of the coefficient, we decompose the coefficient $\phi_{G,k,l}^{(t)}$ to $\psi_{G,k,l}^{(t)}$ and $\varphi_{G,k,l}^{(t)}$.

$$\psi_{G,k,l}^{(t+1)} = \psi_{G,k,l}^{(t)} + \frac{\eta}{n_k}(1-\theta) \cdot \sum_{i=1}^{n_k} \ell_{k,i}'^{(t)} \cdot \sigma_{G,r,k,i}'^{(t)} \mathbf{1}(y_{k,i}=1) \cdot x_{k,i,l} \cdot ||\boldsymbol{\xi}_l||_2^2, \tag{29}$$

$$\varphi_{G,k,l}^{(t+1)} = \varphi_{G,k,l}^{(t)} - \frac{\eta}{n_k}(1-\theta) \cdot \sum_{i=1}^{n_k} \ell_{k,i}'^{(t)} \cdot \sigma_{G,r,k,i}'^{(t)} \mathbf{1}(y_{k,i}=-1) \cdot x_{k,i,l} \cdot ||\boldsymbol{\xi}_l||_2^2, \tag{30}$$

$$\psi_{L,k,l}^{(t+1)} = \psi_{L,k,l}^{(t)} + \frac{\eta}{n_k}\theta \cdot \sum_{i=1}^{n_k} \ell_{k,i}'^{(t)} \cdot \sigma_{G,r,k,i}'^{(t)} \mathbf{1}(y_{k,i}=1) \cdot x_{k,i,l} \cdot ||\boldsymbol{\xi}_l||_2^2, \tag{31}$$

$$\varphi_{L,k,l}^{(t+1)} = \varphi_{L,k,l}^{(t)} - \frac{\eta}{n_k}\theta \cdot \sum_{i=1}^{n_k} \ell_{k,i}'^{(t)} \cdot \sigma_{G,r,k,i}'^{(t)} \mathbf{1}(y_{k,i}=-1) \cdot x_{k,i,l} \cdot ||\boldsymbol{\xi}_l||_2^2. \tag{32}$$

Here, we suppose that $\boldsymbol{\xi}_{k,i} = \sum_{l=1}^{L} x_{k,i,l}\boldsymbol{\xi}_l$, thus we have the following lemma.

**Lemma E.2** ([6]). *Suppose that $\delta \geq 0$ and $L \geq \log(4n/\delta)$, then with probability at least $1-\delta$, we have*

$$\frac{1}{2}\sigma_p^2\sigma_L^2 \leq ||\boldsymbol{\xi}_{k,i}||_2^2 \leq \frac{3}{2}\sigma_p^2\sigma_L^2, \tag{33}$$

$$|\langle \boldsymbol{\xi}_{k,i}, \boldsymbol{\xi}_{k',i'} \rangle| \leq \sigma_p^2 \sqrt{\log(4n^2/\delta)\sigma_L^2}. \tag{34}$$

*where we denote $\sigma_L^2 = \sum_{l=1}^{L} ||\boldsymbol{\xi}_l||_2^2$ as the summation of the norm among all task-irrelevant features.*

### E.3 Generalization analysis

For simplicity, we first introduce the definitions of $F_+$ and $F_-$. Here $F_+$ means the train loss corresponding to the positive class, while $F_-$ means the train loss corresponding to the negative class.

$$F_+(\mathbf{p}) = \sigma(\mathbf{Wp} + \mathbf{Wp}_+) - \sigma(-\mathbf{Wp} + \mathbf{Wp}_+), \tag{35}$$
$$F_-(\mathbf{p}) = \sigma(\mathbf{Wp} + \mathbf{Wp}_-) - \sigma(-\mathbf{Wp} + \mathbf{Wp}_-), \tag{36}$$
$$F(\mathbf{p}) = F_+(\mathbf{p}) - F_-(\mathbf{p}). \tag{37}$$

To illustrate the decoupling of the label from the definitions of $F_+$ and $F_-$, we have the following lemma.

**Lemma E.3.** *Suppose that $y$ is the ground truth label, we have*

$$F_y(\mathbf{p}) - F_{-y}(\mathbf{p}) = y(F_+(\mathbf{p}) - F_-(\mathbf{p})). \tag{38}$$

*Proof.* We consider two situations: $y = +1$ and $y = -1$. If $y = +1$,

$$F_y(\mathbf{p}) - F_{-y}(\mathbf{p}) = F_+(\mathbf{p}) - F_-(\mathbf{p}). \tag{39}$$

If $y = -1$,

$$F_y(\mathbf{p}) - F_{-y}(\mathbf{p}) = F_-(\mathbf{p}) - F_+(\mathbf{p}). \tag{40}$$

In conclusion, we have

$$F_y(\mathbf{p}) - F_{-y}(\mathbf{p}) = y(F_+(\mathbf{p}) - F_-(\mathbf{p})). \tag{41}$$

$\square$

**Lemma E.4.** *Under the modeling of prompt-based federated learning, the expectation of test loss $L_\mathcal{D}$ of an algorithm can be treated as the probability*

$$\mathbb{E}[L_\mathcal{D}] = P(z < 0), z \sim \mathcal{N}(\mu, \sigma^2). \tag{42}$$

*where $\mu$ and $\sigma$ are controlled by the coefficients of feature learning. Thus, the performance of an algorithm can be evaluated by the ratio $\mu/\sigma$.*

*Proof.* Recall that the test error is equivalent to

$$L_\mathcal{D}(\mathbf{p}_L) = P((\langle F_y(\mathbf{p}), \mathbf{x}\rangle - \langle F_{-y}(\mathbf{p}), \mathbf{x}\rangle > 0). \tag{43}$$

According to Lemma E.3, we have that

$$L_\mathcal{D}(\mathbf{p}_L) = P(y\langle F_+(\mathbf{p}) - F_-(\mathbf{p}), \mathbf{x}\rangle > 0). \tag{44}$$

Note that

$$\langle \mathbf{W}, \mathbf{p}\rangle = \begin{bmatrix} \beta\|\mu_G\|_2 \\ \gamma_1\|\mu_1\|_2 \\ \vdots \\ \gamma_K\|\mu_K\|_2 \\ \phi_1\|\xi_1\|_2 \\ \vdots \\ \phi_L\|\xi_L\|_2 \end{bmatrix}. \tag{45}$$

$\mathbf{W}\mathbf{p}_+$ and $\mathbf{W}\mathbf{p}_-$ can be treated as two constant terms. We consider each line of $F(\mathbf{p})$ and $\mathbf{x}$, then the problem is equivalent to

$$y(yf_G(\beta\|\mu_G\|_2) + yf_1(\gamma_1\|\mu_1\|_2) + \sum_j x_j f_{K+j}(\phi_j\|\xi_j\|_2)) \geq 0, \tag{46}$$

where

$$F(\mathbf{p}) = \begin{bmatrix} f_G(\beta\|\mu_G\|_2) \\ f_1(\gamma_1\|\mu_1\|_2) \\ \vdots \\ f_K(\gamma_K\|\mu_K\|_2) \\ f_{K+1}(\phi_1\|\xi_1\|_2) \\ \vdots \\ f_{K+L}(\phi_L\|\xi_L\|_2) \end{bmatrix}. \tag{47}$$

Note that the above equation is equivalent to

$$f_G(\beta\|\mu_G\|_2) + f_1(\gamma_1\|\mu_1\|_2) + \sum_j yx_j f_{K+j}(\phi_j\|\xi_j\|_2)) \geq 0. \tag{48}$$

Note that when we finish training, the coefficients are fixed and thus can be treated as constants. $x_j$ are zero mean Gaussian variables, $y$ are $\{+1, -1\}$ Bernoulli variables and $x_j$ and $y$ are independent. Thus the test error can be defined by two values

$$\mathbb{E}[L_\mathcal{D}] = P(x \geq \frac{\mu}{\sigma}), \tag{49}$$

where

$$\mu = f_G(\beta||\mu_G||_2) + f_1(\gamma_1||\mu_1||_2), \tag{50}$$

$$\sigma = \sum_j f_{K+j}(\phi_j||\xi_j||_2). \tag{51}$$

The problem can be treated as

$$L_{\mathcal{D}} = P(z < 0), z \sim \mathcal{N}(\mu, \sigma^2). \tag{52}$$

$\square$

**Theorem E.5.** *Suppose that the coefficients of CoOp at step $k$ are $\beta_k, \gamma_k$ and $\phi_k$, the coefficients of PromptFL at step $k$ are $\overline{\beta}_k, \overline{\gamma}_k$ and $\overline{\phi}_k$. Here, we define $a := O(\frac{\overline{\beta}_k + \overline{\gamma}_k}{\beta_k + \gamma_k})$ and $b := O(\frac{\overline{\phi}_k}{\phi_k})$ as the ratio of different coefficients and the correlation between two prompts is defined as $\rho \in [0, 1]$. We have that the mixed PromptFolio algorithm has a lower test loss than the mixing test loss of CoOp and PromptFL.*

$$L_{\mathcal{D}}((1-\theta)\mathbf{p}_G + \theta\mathbf{p}_L) \le (1-\theta)L_{\mathcal{D}}(\mathbf{p}_G) + \theta L_{\mathcal{D}}(\mathbf{p}_L), \tag{53}$$

$$\forall\, \theta \in \left[0, \underset{[0,1]}{proj}\left(\frac{C_b - C_c}{2C_a}\right)\right], \quad where \begin{cases} C_a = (b-a)(b^2 + 2\rho b + 1) \\ C_b = (a+b)(b^2 - 1) - 4b(\rho b - 1) \\ C_c = (b-1)\sqrt{(a+b)^2(b+1)^2 - 8ab^2(\rho+1)^2} \end{cases}.$$

*Proof.* Then we suppose that at the end of training, the random variable corresponding to $L_{\mathcal{D}}(\mathbf{p}_G)$ is $z_G \sim \mathcal{N}(\mu_G, \sigma_G^2)$, the random variable corresponding to $L_{\mathcal{D}}(\mathbf{p}_L)$ is $z_L \sim \mathcal{N}(\mu_L, \sigma_L^2)$. Consequently, we have the random variable corresponding to PromptFolio as $z_{GLo} = (1-\theta)z_G + \theta z_L$, then we have

$$z_{GLo} = (1-\theta)z_G + \theta z_L \sim \mathcal{N}((1-\theta)\mu_G + \theta\mu_L, (1-\theta)^2\sigma_G^2 + 2\rho(1-\theta)\theta\sigma_G\sigma L + \theta^2\sigma_L^2) \tag{54}$$

where $0 \le \rho \le 1$ is the correlation coefficient. Assume that $a = \frac{\mu_L}{\mu_G}, b = \frac{\xi_L}{\xi_G}$, we have

$$\frac{(1-\theta)\mu_G + \theta\mu_L}{\sqrt{(1-\theta)^2\sigma_G^2 + 2\rho(1-\theta)\theta\sigma_G\sigma_L + \theta^2\sigma_L^2}} \ge (1-\theta)\frac{\mu_G}{\xi_G} + \theta\frac{\mu_L}{\xi_L} \tag{55}$$

when $a \ge \frac{b^2-b}{b-\rho}$. As a result, we have that

$$L_{\mathcal{D}}((1-\theta)\mathbf{p}_G + \theta\mathbf{p}_L) \le (1-\theta)L_{\mathcal{D}}(\mathbf{p}_G) + \theta L_{\mathcal{D}}(\mathbf{p}_L). \tag{56}$$

Here, we find where this inequality becomes an equality. After further simplification, we observe that this equation is a quartic equation, and we've discarded 0, 1, and another unreasonable solution. The remaining solution can be expressed as

$$\frac{C_b - C_c}{2C_a}, \tag{57}$$

where

$$\begin{aligned} C_a &= (b-a)(b^2 + 2\rho b + 1) \\ C_b &= (a+b)(b^2 - 1) - 4b(\rho b - 1) \\ C_c &= (b-1)\sqrt{(a+b)^2(b+1)^2 - 8ab^2(\rho+1)^2} \end{aligned}. \tag{58}$$

Here, for any $a, b$, we take the

$$\theta \in \left[0, \underset{[0,1]}{proj}\left(\frac{C_b - C_c}{2C_a}\right)\right] \tag{59}$$

due to the derivative of this objective function being less than zero when $\theta = 0$. As a result, when we take $\theta$ in (59), we will obtain the inequality.

$\square$

Additionally, we provide the optimal coefficient of the algorithm and establish the following theorem.

**Theorem E.6.** *Suppose that the coefficients of CoOp at step $k$ are $\beta_k, \gamma_k$ and $\phi_k$, the coefficients of PromptFL at step $k$ are $\overline{\beta}_k, \overline{\gamma}_k$ and $\overline{\phi}_k$. Here, we define $a := O(\frac{\beta_k + \gamma_k}{\overline{\beta}_k + \overline{\gamma}_k})$ and $b := O(\frac{\phi_k}{\overline{\phi}_k})$ as the ratio of different coefficients and the correlation between two prompts is defined as $\rho \in [0, 1]$. Then we have the optimal mixing coefficient $\theta^\star$ as*

$$\theta^\star = \underset{[0,1]}{proj} \left( \frac{a - \rho b}{(a + b^2) - \rho b(a + 1)} \right). \tag{60}$$

*Proof.* Here, we suppose that the local prompt corresponding to random variables $\mu$ and $\sigma$, the global prompt corresponding to the random variables $\overline{\mu}$ and $\overline{\sigma}$. To achieve the highest accuracy, we want to solve the following maximization problem

$$\max_{\theta} \quad \frac{(\theta\mu + (1-\theta)\overline{\mu})^2}{\theta^2\sigma^2 + 2\rho(1-\theta)\theta\sigma\overline{\sigma} + (1-\theta)^2\overline{\sigma}^2} \tag{61}$$

which is equivalent to

$$2(\theta\mu + (1-\theta)\overline{\mu})(\mu - \overline{\mu})(\theta^2\sigma^2 + 2\rho\theta(1-\theta)\sigma\overline{\sigma} + (1-\theta)^2\overline{\sigma}^2)$$
$$= (\theta\mu + (1-\theta)\overline{\mu})(2\theta\sigma^2 + 2\rho(1-2\theta)\sigma\overline{\sigma} - 2(1-\theta)\overline{\sigma}^2)$$
$$(\mu - \overline{\mu})(\theta^2\sigma^2 + 2\rho\theta(1-\theta)\sigma\overline{\sigma} + (1-\theta)^2\overline{\sigma}^2) \tag{62}$$
$$\overset{(a)}{=} (\theta\mu + (1-\theta)\overline{\mu})(\theta\sigma^2 + \rho(1-2\theta)\sigma\overline{\sigma} - \theta(1-\theta)\overline{\sigma}^2).$$

Note that due to $\mu$ and $\overline{\mu}$ are all positive numbers, so we can make a simplification as $(a)$. Here, we take $a = \frac{\mu}{\overline{\mu}}$, $b = \frac{\sigma}{\overline{\sigma}}$, and divide $ab^2$ on both sides, we have

$$(a-1)(\theta^2 b^2 + 2\rho\theta(1-\theta)b + (1-\theta)^2)$$
$$= (\theta a + (1-\theta))(\theta b^2 + \rho(1-2\theta)b - (1-\theta))$$
$$\theta^2 ab^2 + 2\rho\theta(1-\theta)ab + (1-\theta)^2 a - \theta^2 b^2 - 2\rho\theta(1-\theta)b - (1-\theta)^2$$
$$= \theta^2 ab^2 + \rho(1-\theta-\theta)\theta ab - (1-\theta)\theta a + (1-\theta)\theta b^2 + \rho(1-\theta-\theta)(1-\theta)b - (1-\theta)^2. \tag{63}$$

Thus, we take further simplification and we get

$$\rho\theta ab - \theta a + a = \theta b^2 + \rho b - \rho\theta b. \tag{64}$$

Note that this is a linear equation, so we obtain the unique optimal $\theta^\star$

$$\theta^\star = \underset{[0,1]}{proj} \left( \frac{a - \rho b}{(a + b^2) - \rho b(a + 1)} \right). \tag{65}$$

$\square$

# F    Theoretical analysis for PromptFolio

In this section, we provide the whole feature coefficient dynamics through a two-stage analysis. The sections are arranged as follows: We first provide some preliminary lemmas. Then we provide the order of coefficients at the beginning of the training, which is stage one. Finally, we provide the coefficients at the convergence of the loss, which is stage two.

**Remark:** For simplicity of the analysis, we assume that all the clients have the same number of samples and thus $n_1 = \cdots = n_K = n/K$.

## F.1    Coefficient dynamics: stage one

**Lemma F.1.** *Suppose that* $\max(\beta_{G,k}^{(t)}, \beta_{L,k}^{(t)}) = O(1)$, $\max(\gamma_{G,k,k}^{(t)}, \gamma_{L,k,k}^{(t)}) = O(1)$ *and* $\max(\phi_{G,k,k'}^{(t)}, \phi_{L,k,k'}^{(t)}) = O(1)$ *with $i \in [n]$ and $k \in [K]$, for $t \in [0, T_1]$, we have*

$$C_1 \leq \ell_{k,i}^{(t)} \leq 1, \tag{66}$$

*where $C_1$ is a positive constant.*

*Proof.* In the first stage, we consider that the gradient of the loss $\ell'$ is bounded in a constant level. We assume that the coefficients $\gamma^{(t)}, \beta^{(t)}, \phi^{(t)} = O(1)$ for $0 \leq t \leq T_1$. Here we have that the output similarity satisfies:

$$\mathbf{sim}(\mathbf{g}_{k,i}, \mathbf{h}_{k,i}) \leq \max\left\{\sigma(\langle \mathbf{p}_{G,k}^{(t)}, \boldsymbol{\mu}_k\rangle), \sigma(\langle \mathbf{p}_{G,k}^{(t)}, \boldsymbol{\xi}_{k,i}\rangle), \beta_k^{(t)}, \gamma_{k,i}^{(t)}, \phi_{k,i}^{(t)}\right\}$$
$$= O(1). \tag{67}$$

When $t \in [0, T]$, we have

$$\ell_{k,i}'^{(t)} = \frac{1}{1 + \exp(\mathbf{sim}(\mathbf{g}_{k,i}, \mathbf{h}_{k,i}))}$$
$$\geq \frac{1}{1 + O(1)}. \tag{68}$$

$\square$

**Theorem F.2.** *Under Assumption E.1, there exists total number of local updates $T_1 = R_1 E = O(\eta^{-1} K n \sigma_p^2 \sigma_L^2)$ such that*

$$\begin{aligned}
\beta_{G,k}^{(T_1)} &= \Theta((1-\theta)\overline{n}KSNR_G^2) & \beta_{L,k}^{(T_1)} &= \Theta(\theta\overline{n}SNR_G^2), & \forall k \in [K] \\
\gamma_{G,k,k}^{(T_1)} &= \Theta((1-\theta)\overline{n}\chi_k SNR_k^2) & \gamma_{L,k,k}^{(T_1)} &= \Theta(\theta\overline{n}SNR_k^2), & \forall k \in [K] \\
\phi_{G,k,l}^{(T_1)} &= O(1-\theta) & \phi_{L,k,l}^{(T_1)} &= O(\theta), & \forall k \in [K], l \in [L]
\end{aligned} \tag{69}$$

*Proof.* Part 1: Analysis of $\beta_{G,k}^{(T_1)}$ and $\gamma_{G,k}^{(T_1)}$

We first consider the growth of local task-relevant coefficient $\gamma_{G,k,k}^{(t)}$ and $\overline{\gamma}_{G,k}^{(t)}$ on the corresponding client $k$. Consider the iteration equation for the coefficient of signal learning under local gradient descent in the first round, we have:

$$\gamma_{G,k,k}^{(t+1)} = \gamma_{G,k,k}^{(t)} - \frac{\eta}{n_k}(1-\theta) \cdot \sum_{i=1}^{n_k} \ell_{k,i}'^{(t)} \sigma_{G,r,k,i}'^{(t)} ||\boldsymbol{\mu}_k||_2^2. \tag{70}$$

According to Lemma F.1, we have an upper bound for signal learning at the first round of local updates.

$$\gamma_{G,k,k}^{(t+1)} \leq \gamma_{G,k,k}^{(t)} + \eta(1-\theta)||\boldsymbol{\mu}_k||_2^2. \tag{71}$$

Taking a telescoping sum over $t = 0, 1, \ldots, E$, we can obtain the upper bound for signal learning before the first step of weight averaging.

$$\gamma_{G,k,k}^{(E)} \leq \eta(1-\theta)E||\boldsymbol{\mu}_k||_2^2. \tag{72}$$

After taking the average among the coefficients, we have

$$\overline{\gamma}_G^{(E)} \leq \frac{1}{K} \sum_{k'=1}^{K} \frac{\langle \boldsymbol{\mu}_k, \boldsymbol{\mu}_{k'}\rangle}{||\boldsymbol{\mu}_k||_2^2} \eta(1-\theta)E||\boldsymbol{\mu}_k||_2^2. \tag{73}$$

Note that we denote $\chi_k \triangleq \sum_{k'=1}^{K} \frac{\langle \boldsymbol{\mu}_k', \boldsymbol{\mu}_k\rangle}{||\boldsymbol{\mu}_k||}$ as the total similarity between signal vectors among clients. In the following $E$ gradient descent steps on each clients, we have

$$\begin{aligned}
\gamma_{G,k,k}^{(2E)} &\leq \frac{1}{K} \sum_{k'=1}^{K} \frac{\langle \boldsymbol{\mu}_k, \boldsymbol{\mu}_{k'}\rangle}{||\boldsymbol{\mu}_k||_2^2} \eta(1-\theta)E||\boldsymbol{\mu}_k||_2^2 + \eta E||\boldsymbol{\mu}_k||_2^2 \\
&= (\frac{1}{K}\chi_k + 1)\eta(1-\theta)E||\boldsymbol{\mu}_k||_2^2. \tag{74}
\end{aligned}$$

In the second round of update, we apply average among the clients

$$\overline{\gamma}_G^{(2E)} \leq \frac{1}{K}(\chi_k(\frac{\chi_k}{K} + 1) + (K - \chi_k)\frac{\chi_k}{K})\eta(1-\theta)E||\mu_k||_2^2 = \frac{2}{K}\chi_k\eta(1-\theta)E||\mu_k||_2^2. \tag{75}$$

We repeat the computation process and obtain the following results for the third rounds of local updates and the FedAvg process.

$$\gamma_{G,k,k}^{(3E)} \le \frac{2}{K}\chi_k\eta E||\boldsymbol{\mu}_k||_2^2 + \eta E||\mu_k||_2^2 = (\frac{2}{K}\chi_k + 1)\eta(1-\theta)E||\boldsymbol{\mu}_k||_2^2 \tag{76}$$

$$\overline{\gamma}_G^{(3E)} \le \frac{1}{K}(\chi_k(\frac{2}{K}\chi_k + 1) + (K-\chi_k)\frac{2}{K}\chi_k)\eta E||\boldsymbol{\mu}_k||_2^2 = \frac{3}{K}\chi_k\eta(1-\theta)E||\boldsymbol{\mu}_k||_2^2. \tag{77}$$

Thus, after $R_1 = T_1/E$ rounds of communication, the noise memorization has an upper bound:

$$\gamma_{G,k,k}^{(R_1E)} \le (\frac{R_1-1}{K}\chi_k + 1)\eta(1-\theta)E||\boldsymbol{\mu}_k||_2^2 \tag{78}$$

$$\overline{\gamma}_G^{(R_1E)} \le \frac{R_1}{K}\chi_k\eta(1-\theta)E||\boldsymbol{\mu}_k||_2^2. \tag{79}$$

We can similarly obtain the lower bound of this proof. Consider the first round of gradient update, we have

$$\gamma_{G,k,k}^{(t+1)} = \gamma_{G,k,k}^{(t)} + \frac{\eta}{n}(1-\theta)\sum_{i=1}^{n}\ell_{k,i}^{\prime(t)}\sigma_{G,r,k,i}^{\prime(t)}||\boldsymbol{\mu}_k||_2^2$$

$$\overset{(a)}{\ge} \gamma_{G,k,k}^{(t)} + \eta(1-\theta)C_1||\boldsymbol{\mu}_k||_2^2. \tag{80}$$

So before taking the first average, we take a telescope sum over $t = 0, \ldots, E-1$ yielding:

$$\gamma_{G,k,k}^{(E)} \ge \eta(1-\theta)C_1E||\boldsymbol{\mu}_k||_2^2. \tag{81}$$

After the weight average operation, we have

$$\overline{\gamma}_{G,k,k}^{(E)} \ge \frac{1}{K}\sum_{k'=1}^{K}\frac{\langle\boldsymbol{\mu}_k,\boldsymbol{\mu}_{k'}\rangle}{||\boldsymbol{\mu}_k||_2^2}\eta(1-\theta)C_1E||\boldsymbol{\mu}_k||_2^2. \tag{82}$$

Recall that $\chi_k \overset{\Delta}{=} \sum_{k'=1}^{K}\frac{\langle\boldsymbol{\mu}_k,\boldsymbol{\mu}_{k'}\rangle}{||\boldsymbol{\mu}_k||_2^2}$ counts the total similarity between the signal vectors among clients. In the next $E$ gradient descent steps on each client, we have

$$\gamma_{G,k,k}^{(2E)} \ge \frac{\chi_k}{K}\eta(1-\theta)C_1E + \eta C_1E||\boldsymbol{\mu}_k||_2^2$$

$$= (\frac{\chi_k}{K} + 1)\eta(1-\theta)C_1E||\boldsymbol{\mu}_k||_2^2. \tag{83}$$

After gradient descent, we apply the second weight averaging operation on the server and obtain the following result

$$\overline{\gamma}_{G,k,k}^{(2E)} \ge \frac{1}{K}(\chi_k(\frac{\chi_k}{K} + 1) + (K-\chi_k)\frac{\chi_K}{K})\eta(1-\theta)C_1E||\boldsymbol{\mu}_k||_2^2$$

$$= \frac{2}{K}\chi_k\eta(1-\theta)C_1E||\boldsymbol{\mu}_k||_2^2. \tag{84}$$

Similarly, we repeat the computation procedure and obtain the following results for the $R_1$-th round of local updates plus weight average operation:

$$\gamma_{G,k,k}^{(R_1E)} \ge (\frac{R_1-1}{K}\chi_k + 1)\eta(1-\theta)C_1E||\boldsymbol{\mu}_k||_2^2 \tag{85}$$

$$\overline{\gamma}_{G,k}^{(R_1E)} \ge \frac{R_1}{K}\chi_k \cdot \eta(1-\theta)C_1E||\mu_k||_2^2. \tag{86}$$

As a result, we have

$$\overline{\gamma}_{G,k}^{(R_1E)} = \frac{R_1}{K}(1-\theta)E\chi_k\eta C_1||\mu_k||_2^2 = \frac{Cn}{\eta\sigma_q^2 d}\chi_k\eta(1-\theta)C_1||\mu_k||_2^2 = \Theta(\overline{n}(1-\theta)\chi_k\text{SNR}_k^2). \tag{87}$$

Here, the global task-relevant feature follows that $\chi_K = K$, so we have

$$\overline{\beta}_{G,k}^{(R_1E)} = R_1(1-\theta)E\chi_k\eta C_1||\mu_k||_2^2 = \frac{Cn}{\eta\sigma_q^2 d}\chi_k\eta(1-\theta)C_1||\mu_k||_2^2 = \Theta(\overline{n}(1-\theta)K\text{SNR}_G^2). \tag{88}$$

Part 2: Analysis of $\beta_{L,k}^{(T_1)}$ and $\gamma_{L,k,k}^{(T_1)}$

Similar to the global coefficient $\beta_{G,k}^{(T_1)}$ and $\gamma_{G,k,k}^{(T_1)}$, we first analyze $\gamma_{L,k,k}^{(T_1)}$. Consider the iteration equation for the coefficient

$$\gamma_{L,k,k}^{(t+1)} = \gamma_{L,k,k}^{(t)} - \frac{\eta}{n_k}\theta \cdot \sum_{i=1}^{n_k} \ell_{k,i}'^{(t)} \sigma_{L,r,k,i}'^{(t)} ||\boldsymbol{\mu}_k||_2^2. \tag{89}$$

According to Lemma F.1, we have upper bound for $\gamma_{L,k,k}^{(t+1)}$.

$$\gamma_{L,k,k}^{(t+1)} \leq \gamma_{L,k,k}^{(t)} + \eta\theta||\boldsymbol{\mu}_k||_2^2. \tag{90}$$

Then we taking a telescoping sum over $t = 0, 1, \ldots, R_1 E$, we have

$$\gamma_{L,k,k}^{(R_1 E)} \leq \eta\theta R_1 E||\boldsymbol{\mu}_k||_2^2. \tag{91}$$

Similarly, we repeat the computation procedure and have the lower bound of the $\gamma_{L,k,k}^{(t+1)}$

$$\gamma_{L,k,k}^{(t+1)} = \gamma_{L,k,k}^{(t)} - \frac{\eta}{n_k}\theta \cdot \sum_{i=1}^{n_k} \ell_{k,i}'^{(t)} \sigma_{L,r,k,i}'^{(t)} ||\boldsymbol{\mu}_k||_2^2. \tag{92}$$

According to Lemma F.1, we have

$$\gamma_{L,k,k}^{(t+1)} \geq \gamma_{L,k,k}^{(t)} + \eta\theta C_1 ||\boldsymbol{\mu}_k||_2^2. \tag{93}$$

Then we taking a telescoping sum over $t = 0, 1, \ldots, R_1 E$, we have

$$\gamma_{L,k,k}^{(R_1 E)} \geq \eta\theta R_1 C_1 E||\boldsymbol{\mu}_k||_2^2. \tag{94}$$

As a result, we have

$$\overline{\gamma}_{L,k,k}^{(R_1 E)} = R_1 \theta E \eta C_1 ||\mu_k||_2^2 = \frac{Cn}{\eta \sigma_q^2 d}\chi_k \eta(1-\theta)C_1||\mu_k||_2^2 = \Theta(\overline{n}\theta\mathrm{SNR}_k^2). \tag{95}$$

The global feature and local feature have the same weight and take the same order.

$$\overline{\beta}_{L,k,k}^{(R_1 E)} = R_1 \theta E \chi_k \eta C_1 ||\mu_k||_2^2 = \frac{Cn}{\eta \sigma_q^2 d}\chi_k \eta(1-\theta)C_1||\mu_k||_2^2 = \Theta(\overline{n}\theta\mathrm{SNR}_G^2). \tag{96}$$

Part 3: Analysis of $\phi_{G,k,l}^{(T_1)}$ and $\phi_{L,k,l}^{(T_1)}$

We first establish the lower bound for $\phi_{G,k,l}^{(T_1)}$. We show that

$$\langle \mathbf{p}_{G,k}^{(t)}, \boldsymbol{\xi}_{k,i} \rangle = \langle \mathbf{p}_{G,k}^{(0)}, \boldsymbol{\xi}_{k,i} \rangle + \sum_{i'=1}^{n} (\psi_{G,k,l}^{(t)} + \varphi_{G,k,l}^{(t)})||\boldsymbol{\xi}_{k,i'}||_2^{-2}\langle \boldsymbol{\xi}_{G,k,i}, \boldsymbol{\xi}_{G,k,i'} \rangle \tag{97}$$

$$\overset{(a)}{\geq} \langle \mathbf{p}_{G,k}^{(0)}, \boldsymbol{\xi}_{k,i} \rangle + \psi_{G,k,l}^{(t)} - 2\sqrt{\frac{\log(4n^2/\delta)}{\sigma_L^2}}\sum_{i'\neq i}(|\psi_{G,k,l}^{(t)}| + |\varphi_{G,k,l}^{(t)}|) \tag{98}$$

$$\overset{(b)}{\geq} \langle \mathbf{p}_{G,k}^{(0)}, \boldsymbol{\xi}_{k,i} \rangle + \psi_{G,k,l}^{(t)} - 4C_2 n\sqrt{\frac{\log(4n^2/\delta)}{\sigma_L^2}}, \tag{99}$$

where $C_2$ is a positive constant that satisfies $C_2 \geq \max\{\psi_{G,k,l}^{(t)}, \varphi_{G,k,l}^{(t)}\}$, $(a)$ is by Lemma E.2 and $(b)$ is by definition of $C_2$.

Here, we suppose that $\Phi_{G,k,l}^{(t)} = \max\left\{\langle \mathbf{p}_{G,k}^{(0)}, \boldsymbol{\xi}_{k,i} \rangle + \psi_{G,k,l}^{(t)} - 4C_2 n\sqrt{\frac{\log(4n^2/\delta)}{\sigma_L^2}}\right\}$. At initialization, it is easy to check that:

$$\Phi_{G,k,l}^{(0)} \geq \frac{1}{4}\sigma_0\sigma_p\sigma_L - 4C_2 n\sqrt{\frac{\log(4n^2/\delta)}{\sigma_L^2}} \overset{(a)}{\geq} 0, \tag{100}$$

where $(a)$ is by the following condition:

$$\sigma_0 \geq C_3 n \frac{\sqrt{\log(4n^2/\delta)}}{\sigma_p \sigma_L^2}. \tag{101}$$

We can compute the growth of $\Phi_{G,k,l}^{(t)}$ as follows:

$$\Phi_{G,k,l}^{(t+1)} = \Phi_{G,k,l}^{(t)} + \frac{\eta}{n}(1-\theta)\ell_{k,i}^{\prime(t)}\sigma_{G,r,k,i}^{\prime(t)}\|\boldsymbol{\xi}_{k,i}\|_2^2$$

$$\overset{(a)}{\geq} \Phi_{G,k,l}^{(t)} + \frac{\eta}{2n}(1-\theta)C_1\sigma_p^2\sigma_L^2. \tag{102}$$

Before the first step of weight average, we take the telescoping sum:

$$\Phi_{G,k,l}^{(E)} \geq \frac{\eta E C_1}{2n}(1-\theta)\sigma_p^2\sigma_L^2. \tag{103}$$

Then, we perform weight average operation

$$\overline{\Phi}_{G,l}^{(E)} \geq \frac{1}{K}\frac{\eta E C_1}{2n}(1-\theta)\sigma_p^2\sigma_L^2. \tag{104}$$

The next $E$ gradient descent steps yield:

$$\Phi_{G,k,l}^{(2E)} \geq \frac{K+1}{K}\frac{\eta E C_1}{2n}(1-\theta)\sigma_p^2\sigma_L^2. \tag{105}$$

Here, we take the average among the clients and we get:

$$\overline{\Phi}_{G,l}^{(2E)} \geq (\frac{1}{K}\frac{K+1}{K} + \frac{K-1}{K}\frac{1}{K})\frac{\eta E C_1}{2n}(1-\theta)\sigma_p^2\sigma_L^2 \geq \frac{2}{K}\frac{\eta E C_1}{2n}(1-\theta)\sigma_p^2\sigma_L^2. \tag{106}$$

We use the same technique to obtain the lower bound of noise memorization

$$\overline{\Phi}_{G,l}^{(R_1 E)} \geq \frac{R_1}{K}\frac{\eta E C_1}{2n}(1-\theta)\sigma_p^2\sigma_L^2. \tag{107}$$

Finally, we confirm that

$$\psi_{G,k,l}^{(t)} \geq \frac{\eta T_1 C_1}{2nK}(1-\theta)\sigma_p^2\sigma_L^2 - \langle \mathbf{p}_{G,k}^{(0)}, \boldsymbol{\xi}_{k,i}\rangle + 4C_2 n\sqrt{\frac{\log(4n^2/\delta)}{\sigma_L^2}} \tag{108}$$

$$\overset{(a)}{\geq} \frac{\eta T_1 C_1}{2nK}(1-\theta)\sigma_p^2\sigma_L^2 - \langle \mathbf{p}_{G,k}^{(0)}, \boldsymbol{\xi}_{k,i}\rangle + C_3 \tag{109}$$

$$\overset{(b)}{\geq} C_4, \tag{110}$$

where the inequality $(a)$ is by definition of constant $C_3 \leq 4C_2 n\sqrt{\frac{\log(4n^2/\delta)}{\sigma_L^2}}$ which holds when $\sigma_L^2 \geq C_5 \log(4n^2/\delta)n^2$ and the inequality $(b)$ is by taking the value of $T_1$.

Then we provide the upper bound of the $\phi_{G,k,l}^{(T_1)}$. Here, we suppose that $\Psi_{G,k,l}^{(t)} = \max\left\{\psi_{G,k,l}^{(t)}, -\varphi_{G,k,l}^{(t)}\right\}$. Similarly, we define the coefficient after weight average $\overline{\Psi}_{G,k,l}^{(t)} = \max\left\{\overline{\psi}_{G,k,l}^{(t)}, -\overline{\varphi}_{G,k,l}^{(t)}\right\}$. Clearly, we have that $\overline{\Psi}_{G,k,l}^{(0)} = 0$ for all $l \in [L]$ and $k \in [K]$ by definition. Then we have

$$\Psi_{G,k,l}^{(t+1)} = \Psi_{G,k,l}^{(t)} + \frac{\eta}{n}(1-\theta)\ell_{k,i}^{\prime(t)}\sigma_{G,r,k,i}^{\prime(t)}\|\boldsymbol{\xi}_{k,i}\|_2^2$$

$$\overset{(a)}{\leq} \Psi_{G,k,l}^{(t)} + \frac{\eta}{2n}(1-\theta)\sigma_p^2\sigma_L^2, \tag{111}$$

where $(a)$ follows $|\ell_{k,i}^{\prime(t)}| \leq 1$ and $\sigma' \leq 1$ in Lemma F.1 and Lemma E.2. Similar to above proof, we have that

$$\Psi_{G,k,l}^{(E)} \leq \frac{\eta E}{2n}(1-\theta)\sigma_p^2\sigma_L^2. \tag{112}$$

After $t = E$ steps, we perform a weight average operation and we have

$$\overline{\Psi}_{G,k,l}^{(E)} \leq \frac{1}{K}\frac{\eta E}{2n}(1-\theta)\sigma_p^2\sigma_L^2. \tag{113}$$

Similar to the above proof, we have

$$\overline{\Psi}_{G,k,l}^{(R_1 E)} \leq \frac{R_1}{K}\frac{\eta E}{2n}(1-\theta)\sigma_p^2\sigma_L^2. \tag{114}$$

Thus, it is confirmed that $\Psi_{G,k,l}^{(T_1)} = O(1-\theta)$. Similar to the convergence of $\Psi_{G,k,l}^{(T_1)} = O(1-\theta)$, we have that $\Psi_{L,k,l}^{(T_1)} = O(\theta)$. $\qquad\square$

**Lemma F.3.** *There exists total number of local updates $T_1 = R_1 E = O(\eta^{-1}Kn\sigma_p^2\sigma_L^2)$ such that*

$$\overline{\beta}^{(T_1)} = \Theta(\overline{n}K SNR_G^2), \quad \overline{\gamma}_k^{(T_1)} = \Theta(\overline{n}\chi_k SNR_k^2), \quad \overline{\phi}_l^{(T_1)} = O(1) \quad \forall k \in [K], l \in [L]. \tag{115}$$

*Here, $\overline{n} = \sum_k n_k/K$ is the average number of data in each client and $SNR_G = ||\boldsymbol{\mu}_G||/(\sigma_p\sqrt{m})$, $SNR_k = ||\boldsymbol{\mu}_k||/(\sigma_p\sqrt{m})$ denote the signal-noise ratio between the task-relevant feature and task-irrelevant feature. Here, we define $\chi_k = \sum_{k'=1}^K \langle \boldsymbol{\mu}_k, \boldsymbol{\mu}_{k'} \rangle /||\boldsymbol{\mu}_k||_2^2$.*

*Proof.* The proof of the result follows the proof of global coefficient in Lemma F.3. We take $\theta = 0$ to obtain the final result. $\qquad\square$

## F.2 Coefficient dynamics: stage two

In this stage, we focus on the proof of scaling behaviour for the coefficients of features $\beta_G, \gamma_G, \beta_L, \gamma_L$. We first provide the following lemma.

**Lemma F.4.** *Under Assumption E.1, we have*

$$\begin{aligned}
\langle \mathbf{p}_{G,k}^{(t)}, y\boldsymbol{\mu}_G \rangle &= \langle \mathbf{p}_{G,k}^{(0)}, y\boldsymbol{\mu}_G \rangle + \overline{\beta}_G^{(t)}, \quad \langle \mathbf{p}_{G,k}^{(t)}, y\boldsymbol{\mu}_k \rangle = \langle \mathbf{p}_{G,k}^{(0)}, y\boldsymbol{\mu}_k \rangle + \overline{\gamma}_{G,k}^{(t)}, & \forall k \in [K] \\
\langle \mathbf{p}_{L,k}^{(t)}, y\boldsymbol{\mu}_L \rangle &= \langle \mathbf{p}_{L,k}^{(0)}, y\boldsymbol{\mu}_L \rangle + \overline{\beta}_G^{(t)}, \quad \langle \mathbf{p}_{L,k}^{(t)}, y\boldsymbol{\mu}_k \rangle = \langle \mathbf{p}_{L,k}^{(0)}, y\boldsymbol{\mu}_k \rangle + \overline{\gamma}_{L,k}^{(t)}, & \forall k \in [K]
\end{aligned} \tag{116}$$

*for all $k \in [K]$.*

*Proof.* According to the update formula of $\mathbf{p}_G$ and $\mathbf{p}_L$, we have

$$\begin{aligned}
\langle \overline{\mathbf{p}}_G^{(t)} - \overline{\mathbf{p}}_G^{(0)}, \boldsymbol{\mu}_G \rangle &= y\overline{\gamma}_G^{(t)} + \sum_{k=1}^K \gamma_{G,k}^{(t)}||\boldsymbol{\mu}_k||_2^{-2}\langle \boldsymbol{\mu}_k, \boldsymbol{\mu}_G \rangle + \sum_{l=1}^L \overline{\phi}_{G,l}^{(t)}||\boldsymbol{\xi}_l||_2^{-2}\boldsymbol{\xi}_l \\
&= y\overline{\gamma}_G^{(t)}, \tag{117}
\end{aligned}$$

where the second equation is by the orthogonal assumption between the feature vector and noise vector. Here, these equations follow the same proof and we thus complete the proof. $\qquad\square$

**Lemma F.5.** *Under Assumption E.1, for $0 \leq t \leq T^*$, where $T^* = \eta^{-1}poly(||\boldsymbol{\mu}||_2^{-1}, \sigma_L^{-2}\sigma_p^{-2}, \sigma_0^{-1}, n, m)$, we prove that*

$$0 \leq \overline{\beta}_G^{(t)} \leq \beta_{G,k}^{(t)} \leq (1-\theta)\overline{n}K SNR_G^2 \log(T^*), \tag{118}$$

$$0 \leq \overline{\gamma}_{G,k}^{(t)} \leq \gamma_{G,k,k}^{(t)} \leq (1-\theta)\overline{n}\chi_k SNR_k^2 \log(T^*), \tag{119}$$

$$0 \leq \overline{\beta}_L^{(t)} \leq \beta_{L,k}^{(t)} \leq \theta\overline{n}K SNR_G^2 \log(T^*), \tag{120}$$

$$0 \leq \overline{\gamma}_{L,k}^{(t)} \leq \gamma_{L,k,k}^{(t)} \leq \theta\overline{n}K SNR_k^2 \log(T^*), \tag{121}$$

*for all $k \in [K]$.*

*Proof.* Here, we only prove that $0 \leq \overline{\gamma}_{G,k}^{(t)} \leq \gamma_{G,k,k}^{(t)} \leq (1-\theta)\overline{n}\chi_k \text{SNR}_k^2 \log(T^*)$. The proof of other coefficients can be generalized from this proof. Considering the update formula, we have

$$\overline{\gamma}_G^{(t+E)} = \overline{\gamma}_G^{(t)} + \sum_{\tau=1}^{E} \frac{(1-\theta)\eta\chi_k}{nK} \sum_{i=1}^{n} \ell_{k,i}'^{(t+\tau)} \sigma_{G,r,k,i} ||\boldsymbol{\mu}_k||_2^2. \tag{122}$$

Let $T_b = R_b E$ to be the last time $t \leq T^*$ that $\overline{\gamma}_{G,k}^{(t)} \leq 0.5 \log(T^*)\overline{n}\text{SNR}_k^2$. We have the lower bound and upper bound of $\langle \mathbf{p}_{G,k}^{(t)}, \mu_k \rangle$:

$$0.25 n\chi_k \text{SNR}_k^2 \log(T^*) \leq \langle \mathbf{p}_{G,k}^{(t)}, \mu_k \rangle \leq 0.5 n\chi_k \text{SNR}_k^2 \log(T^*). \tag{123}$$

Due to that

$$\langle \mathbf{p}_{G,k}^{(t)}, \mu_k \rangle \overset{(a)}{=} \langle \mathbf{p}_{G,k}^{(0)}, \mu_k \rangle + \overline{\gamma}_{G,k}^{(t)} \tag{124}$$

$$\overset{(b)}{\geq} -0.5(1-\theta)\alpha + 0.5(1-\theta)n\chi_k \text{SNR}_k^2 \log(T^*) \tag{125}$$

$$\overset{(c)}{\geq} 0.25(1-\theta)n\chi_k \text{SNR}_k^2 \log(T^*), \tag{126}$$

where the first equality $(a)$ is by the update formula of $\gamma$, the second inequality $(b)$ is by $\overline{\gamma}_{G,k} \geq 0.5(1-\theta)n\chi_k \text{SNR}_k^2 \log(T^*)$ and $\langle \mathbf{p}_{G,k}^{(0)}, \mu_k \rangle \geq -0.5(1-\theta)\alpha$ due to the definition of $T_c$ and $\alpha$. The last inequality $(c)$ is by $\alpha \leq 0.5 n\chi_k \text{SNR}_k^2 \log(T^*)$. Similarly, we can also derive the upper bound as follows:

$$\langle \mathbf{p}_{G,k}^{(t)}, \mu_k \rangle \overset{(a)}{=} \langle \mathbf{p}_{G,k}^{(0)}, \mu_k \rangle + \overline{\gamma}_{G,k}^{(t)} \tag{127}$$

$$\overset{(b)}{\leq} 0.5(1-\theta)\alpha + 0.5(1-\theta)n\chi_k \text{SNR}_k^2 \log(T^*) \tag{128}$$

$$\overset{(c)}{\leq} 0.5(1-\theta)n\chi_k \text{SNR}_k^2 \log(T^*). \tag{129}$$

According to the above formula, we have

$$\overline{\gamma}_{G,k}^{(T_a)} = \overline{\gamma}_{G,k}^{(T_b)} + \frac{\eta\chi_k}{nK} \sum_{\tau=1}^{E} \sum_{i=1}^{n} \ell_{k,i}'^{(T_c+\tau)} \sigma'(\langle \mathbf{p}_{G,k+1}^{(T_c+\tau)}, y_{k,i}\mu_k \rangle)||\boldsymbol{\mu}_k||_2^2$$

$$+ \sum_{R_b < R < R_a} \frac{\eta\chi_k}{nK} \sum_{\tau=1}^{E} \sum_{i=1}^{n} \ell_{k,i}'^{(RE+\tau)} \sigma'(\langle \mathbf{p}_{G,k+1}^{(RE+\tau)}, y_{k,i}\mu_k \rangle)||\boldsymbol{\mu}_k||_2^2 \tag{130}$$

$$\overset{(a)}{\leq} \overline{\gamma}_{G,k}^{(T_b)} + \frac{\eta\chi_k}{nK} \sum_{\tau=1}^{E} \sum_{i=1}^{n} \ell_{k,i}'^{(T_c+\tau)} \sigma'(\langle \mathbf{p}_{G,k+1}^{(T_c+\tau)}, y_{k,i}\mu_k \rangle)||\boldsymbol{\mu}_k||_2^2$$

$$+ \sum_{R_b < R < R_a} \frac{\eta\chi_k}{nK} \sum_{\tau=1}^{E} \exp(1 - \sigma(\langle \mathbf{p}_{G,k+1}^{(RE+\tau)}, y_{k,i}\mu_k \rangle))\sigma'(\langle \mathbf{p}_{G,k+1}^{(RE+\tau)}, y_{k,i}\mu_k \rangle)||\boldsymbol{\mu}_k||_2^2 \tag{131}$$

$$\overset{(b)}{\leq} \overline{\gamma}_{G,k}^{(T_b)} + 0.25(1-\theta)n\chi_k \text{SNR}_k^2 \log(T^*)$$

$$+ 0.25 T^* \exp(-\log(T^*)n\chi_k \text{SNR}_k^2) \log(T^*)n\chi_k \text{SNR}_k^2 \tag{132}$$

$$\overset{(c)}{\leq} (1-\theta)n\chi_k \text{SNR}_k^2 \log(T^*), \tag{133}$$

where the first inequality $(a)$ is by the definition of $\ell'$, the second inequality $(b)$ is by the above formula and the third inequality $(c)$ is due to that $n\chi_k \text{SNR}_k^2 \geq 1$ and the induction hypothesis $\gamma_{G,k}^{(T_b)} \leq 0.5(1-\theta)n\chi_k \text{SNR}_k^2 \log(T^*)$. $\qquad\square$

