# OpenReview forum: "Federated Learning from Vision-Language Foundation Models: Theoretical Analysis and Method"
_NeurIPS.cc/2024/Conference — NeurIPS 2024 poster_

### Official Review · Reviewer_YEHg · 2024-06-23

**Soundness:** 2
**Presentation:** 3
**Contribution:** 2
**Rating:** 5
**Confidence:** 4

**Summary:**

In this papaer, the authors proposed a method to finetune vision-language model (CLIP) via prompt learning. Specifically, the author optimize a global textual prompt which will be optimized via FedAvg, while each client additionally optimize a local textual prompt. The proposed method is a combination of the existing methods CoOp and PromptFL and outperforms both methods.

**Strengths:**

1. The paper is well-written.
2. The theoretical analysis of the proposed method is thorough and clear.

**Weaknesses:**

1. The proposed method is basically a combination of both CoOp and PromptFL, or to be more specific, optimizing both client-specific and client-agnostic textual prompt. The novelty of the proposed needs to be further highlighted in the main paper.
2. The combination strategy seems to be a bit straight forward, i.e., using a single scalar value $\theta$, the author should discuss other possibilities.
3. The authors should compare with other prompt-based tuning methods for FL, e.g., [1][2]. Also, adding more ablation studies could be beneficial.

[1] Guo T, Guo S, Wang J. Pfedprompt: Learning personalized prompt for vision-language models in federated learning[C]//Proceedings of the ACM Web Conference 2023. 2023: 1364-1374.

[2] Qiu C, Li X, Mummadi C K, et al. Text-driven Prompt Generation for Vision-Language Models in Federated Learning[J]. arXiv preprint arXiv:2310.06123, 2023.

**Questions:**

Please see the questions listed in the weakneses section.

**Limitations:**

The authors do not discuss the limitation of the method.

---

> ### Author Rebuttal · Authors · 2024-08-07
>
> Thank you for your feedback and for mentioning that our theoretical analysis is "thorough and clear."
>
> > Weakness 1: The proposed method is basically a combination of both CoOp and PromptFL, or to be more specific, optimizing both client-specific and client-agnostic textual prompt. The novelty of the proposed needs to be further highlighted in the main paper.
>
> We wish to emphasize that our paper mainly focuses on developing **a theoretical framework for federated prompt learning**. Our theoretical framework introduces feature learning theory, decomposing the learnable prompt into task-relevant and task-irrelevant features. By a thorough analysis of the dynamics of feature coefficients, we demonstrate the **convergence and generalization of federated prompt learning in Theorem C.5 and Theorem D.5** by evaluating the ratio between the coefficients of task-relevant and task-irrelevant features.
>
> Under this theoretical framework, we further construct the portfolio optimization method for federated prompt learning, i.e., PromptFolio, and **derive an analytical solution based on this theoretical framework**. PromptFolio seeks to balance the global consensus (PromptFL) and local personalization (CoOp) of federated prompt learning. This simple but efficient method can even outperform many SOTA federated prompt learning methods, such as PromptFL, PromptFL+FedProx, PromptFL+FedAMP, FedTPG, etc. The novelty will be highlighted in the final version.
>
> > Weakness 2: The combination strategy seems to be a bit straight forward, i.e., using a single scalar value $\theta$, the author should discuss other possibilities.
>
> **The combination strategy:** We would like to clarify the portfolio optimization is not a simple combination. Instead, this is an **analytical solution under the feature learning theoretical framework**. Our feature learning framework evaluates the performance of prompt fine-tuning by considering the ratio of task-relevant to task-irrelevant factors. We draw an analogy with portfolio optimization, where the precise fine-tuning of the combination ratios of various assets can maintain returns while mitigating risk. By employing this simple yet effective algorithm, we demonstrate the practical application of our theoretical framework. It's important to note that in portfolio optimization, **adjusting the scalar value of each asset significantly impacts the performance**, making the fine-tuning of mixing ratios and the determination of optimal coefficients a non-trivial task.
>
> **Other possibilities:** Also, our theoretical framework is flexible and can be **extended to incorporate more complex combination strategies**. For example, some optimal transport-based algorithms can be a possible way to construct the combination of global and local prompts.
>
> > Weakness 3: The authors should compare with other prompt-based tuning methods for FL, e.g., [1][2]. Also, adding more ablation studies could be beneficial.
>
> **Further comparisons**: We've added the comparison results between our PromptFolio algorithm and FedTPG [2]. The results are listed in **Table R1 in the PDF** in the global response. Since the code for pFedPrompt [1] is not available online, we were unable to include a comparison with it due to time limitations. It is worth mentioning that our algorithm demonstrates good performance.
>
> **More ablation studies**: Additionally, we have included ablation studies on more users, different shot numbers, and different backbones. The further experimental results can also be referred to in the PDF in the global response.
>
> > Limitations: The authors do not discuss the limitation of the method.
>
> Clarification: Actually, **the limitation was discussed at the end of the Conclusion**. We mentioned that the limitation of this work lies in the use of a simplified text encoder with a single activation function. We hope that the development of theoretical tools will provide future work for more practical and complex models to capture the behaviors of foundation models.

---

> > ### Comment · Reviewer_YEHg · 2024-08-12
> >
> > Thanks for the rebuttal. I have raised my score to Borderline Accept.

---

> > > ### Author Response · Authors · 2024-08-13
> > >
> > > Thanks for reconsidering our work and raising your score. We appreciate your constructive feedback and supports.

---

### Official Review · Reviewer_kpWj · 2024-07-12

**Soundness:** 3
**Presentation:** 2
**Contribution:** 3
**Rating:** 6
**Confidence:** 2

**Summary:**

This paper discusses the integration of pretrained vision-language foundation models, such as CLIP, into federated learning. The idea is to use prompt-based federated learning to minimize the communication and computational costs. The authors go into the theoretical analysis to understand the performance of this approach

**Strengths:**

- The authors provide a theoretical understanding on prompt Federated Learning for models like CLIP.
- The results demonstrate that the combination of global and local prompts can support increased accuracies.
- The overall investigation of this space is interesting and useful for the community

**Weaknesses:**

- While the overall approach is quite interesting, the authors restrict their investigation in a much simpler classification task. It is unclear why a larger foundation model is required for simpler classification tasks, where traditional FL techniques on vanilla models (e.g., reset, mobileNet etc) might work better towards learning new unseen distributions that a foundation model might not have seen (e.g,. x-ray hospital images etc).
- Following the previous comment, it is unclear how this method would work on unseen distributions that most FL applications are mostly useful for. Most FL models are quite good at generalising to data that can be found on the public internet, but it is unknown how well they would work on out-of-distribution private data.
- Given the fact that there are a few related prompt-based FL methods (e.g., promtFL, FedPrompt, etc) limit a bit the novelty of this work to mostly providing a solid theoretical analysis.
- There might be some privacy implications from this work, as sensitive data are shared.
- The evaluation was done with mostly 10 (up to 50) users, whereas in typical FL scenarios we want to learn from million of users. It is unclear how these results would generalise. Overall, the evaluation is a bit simplistic, showing only accuracy results on a few tasks.

**Questions:**

See above

**Limitations:**

See above

---

> ### Author Rebuttal · Authors · 2024-08-07
>
> Thank you for your appreciation of our approach and consider that our paper is "useful for the community". Below, we address each of your points in detail.
>
> > Q1(1): While the overall approach is quite interesting, the authors restrict their investigation in a much simpler classification task.
>
> Firstly, it is worth mentioning that our paper mainly focuses on proposing the **first theoretical framework for federated prompt learning with feature learning theory**. We use the classification task as an example due to the following reasons:
>
> - Most prompt fine-tuning under VLM, as demonstrated in CoOp and PromptFL [40, 16], **only conduct experiments on classification tasks**.
> - Also, in the feature learning community, the analysis mainly **focused on classification tasks** [1, 6, 7, 18].
> - Most federated learning on VLM research [15, 37, 27] **only conduct experiments on classification tasks**.
>
> From these aspects, we use a classification task to demonstrate the application of our theoretical framework in this paper.
>
>
> > Q1(2): It is unclear why a larger foundation model is required for simpler classification tasks, where traditional FL techniques on vanilla models (e.g., reset, mobileNet etc) might work better towards learning new unseen distributions that a foundation model might not have seen (e.g,. x-ray hospital images etc).
>
> To address the first question, we further clarify why we perfer foundation models with prompt learning rather than vanilla models in federated learning.
> - Foundation models with prompt learning **address the concern of communication cost** in federated learning. In a general prompt learning setting, only **8k** learnable parameters are used, which is a **very small scale**, while the performance of foundation models can be maintained and observed at a high level. Here, large foundation models will not increase the boundary of communication but will actually decrease it.
> - Prompt learning can **address the privacy concern** during federated learning. It is worth mentioning that the prompt here is a sequence of dense vectors sent to the text encoder with only 8k parameters. This small number of parameters make it **difficult to recover the original data information**. In our proposed method, the prompts that encapsulate information in a highly abstract form are sent to the server while **the sensitive data are kept on the local client**.
>
>
> > Q2: Following the previous comment, it is unclear how this method would work on unseen distributions that most FL applications are mostly useful for. Most FL models are quite good at generalising to data that can be found on the public internet, but it is unknown how well they would work on out-of-distribution private data.
>
> Thank you for these comments. **Our federated prompt learning method has good generalization ability inherently because of the integration of the large vision-language foundation models.** This has been verified in some recent works, such as **FedTPG** [a] and **FedGPG** [12] that are published at ICLR 2024 and ICML 2024, respectively. These two works both show that federated prompt learning benefits from the **generalizable representation of the foundation models** and has very **robust performance** in various out-of-distribution (OOD) scenarios, such as local, base, and novel settings.
>
> Since our paper aims to develop the first theoretical framework for federated prompt learning, the experiments of OOD scenarios are out of scope.
>
> [a] Qiu C, Li X, Mummadi C K, et al. Text-driven Prompt Generation for Vision-Language Models in Federated Learning, ICLR 2024.
>
> > Q3: Given the fact that there are a few related prompt-based FL methods (e.g., promtFL, FedPrompt, etc) limit a bit the novelty of this work to mostly providing a solid theoretical analysis.
>
> Thank you for pointing out this aspect. We'd like to mention that our work mainly **focuses on proposing the first theoretical framework**. This theoretical framework introduces feature learning theory, decomposing the learnable prompts into task-relevant and task-irrelevant features. By a thorough analysis of the dynamics of feature coefficients, we demonstrate the **convergence and generalization of federated prompt learning in Theorem C.5 and Theorem D.5** by evaluating the ratio between the coefficients of task-relevant and task-irrelevant features. Besides, we leverage the **idea of portfolio optimization to derive our PromptFolio algorithm**. This simple but efficient algorithm exemplifies the practicality of our analytical framework, which can be extended to further prompt-based FL methods.
>
> > Q4: There might be some privacy implications from this work, as sensitive data are shared.
>
> Prompt learning doesn't have privacy issue. Please refer to more analysis in the answer of Q1(2).
>
> > Q5: The evaluation was done with mostly 10 (up to 50) users, whereas in typical FL scenarios we want to learn from million of users. It is unclear how these results would generalise. Overall, the evaluation is a bit simplistic, showing only accuracy results on a few tasks.
>
> Thank you for your valuable comment. We need to mention that we have not seen any federated learning papers that conduct experiments with millions of users. Instead, most research papers use fewer than 100 users due to dataset limitations. During the rebuttal, we have added a new experiment with 100 users and presented the result in **Figure R3 in the global response PDF**. The experimental results with 100 users are consistent with the previous results.
>
> Besides, our paper mainly focuses on providing a theoretical feature learning framework for federated prompt learning. **As a theoretical paper, 100 users are large enough to demonstrate the practicality of our theoretical framework**.

---

> > ### Comment · Reviewer_kpWj · 2024-08-12
> >
> > I would like to thank the authors for clarifying multiple points and overall providing a detailed rebuttal to all authors. After reading all reviews and comments, I think I will stay with my weak accept recommendation.

---

> > > ### Author Response · Authors · 2024-08-13
> > >
> > > Thanks for your positive feedbacks and consistent supports for our work.

---

### Official Review · Reviewer_XM5m · 2024-07-12

**Soundness:** 2
**Presentation:** 2
**Contribution:** 2
**Rating:** 4
**Confidence:** 5

**Summary:**

This paper studies theoretical properties of prompt based federated learning methods for visual language models. Following the proposed theoretical results, a new algorithm named Global-local Prompt Portfolio for Federated Learning (PromptFolio) was proposed. The proposed algorithm was examined in image classification tasks using a CLIP model.

**Strengths:**

Analyzing prompt-based federated learning methods with feature representations and exploiting their relation to portfolio optimization is a nice idea. The proposed analyses also led to a new algorithm for prompt based federated learning.

**Weaknesses:**

Experimental analyses are limited and should be improved as well.

**Questions:**

In the algorithm, each client sends prompts to a server. How do assure privacy protection in this case?

In the analyses, CLIP was utilized as a foundation model. However, the main claim of the paper is providing a generic framework for a more general category of visual language foundation models. Therefore, the proposed method should be examined using the other visual language models as well.

Moreover, CLIP was used for image classification tasks, with limited prompts. In order to show the generalization of the proposed method for prompts with different structures, additional results should be given for the other tasks, such as image captioning and scene description, as well.

In Fig. 2.b, the best accuracy is achieved for 20 users, and the lowest accuracy is achieved for 5 users. Could you please elaborate this result? Could you please provide additional analyses for a larger number of users as well?

**Limitations:**

Limitations were partially addressed.

---

> ### Author Rebuttal · Authors · 2024-08-07
>
> We greatly appreciate your feedback and inquiries. We're thankful that you consider our examination of the connection between federated prompt learning and portfolio optimization to be "a nice idea".
>
> > Question 1: In the algorithm, each client sends prompts to a server. How do assure privacy protection in this case?
>
> Thank you for pointing out this concern. Actually, prompt learning can protect privacy during federated learning. It is worth mentioning that the prompt here is a **sequence of dense vectors** sent to the text encoder with only 8K parameters. This small number of parameters make it **difficult to recover the original data information**. In our proposed method, the prompts that encapsulate information in a **highly abstract form** are sent to the server while the sensitive data are kept on the local client. Thus, there are no privacy concerns in prompt learning.
>
> > Question 2: In the analyses, CLIP was utilized as a foundation model. However, the main claim of the paper is providing a generic framework for a more general category of visual language foundation models. Therefore, the proposed method should be examined using the other visual language models as well.
>
> We appreciate the reviewer's feedback. Since we aim to propose a theoretical analysis framework for prompt tuning, we **used CLIP as an example**. To demonstrate that our model is not limited to a specific structure, we have **replaced the ViT backbone with ResNet** in our experiments. The results are shown in **Figure R2** in the PDF. It's observed that the results with ResNet **also confirm our theoretical analysis**. Furthermore, we plan to include experiments with additional visual language models such as ALIGN in our future work.
>
> > Question 3: Moreover, CLIP was used for image classification tasks, with limited prompts. In order to show the generalization of the proposed method for prompts with different structures, additional results should be given for the other tasks, such as image captioning and scene description, as well.
>
> Thank you for your suggestion. Firstly, it is worth mentioning that our paper mainly focuses on proposing the **first theoretical framework for federated prompt learning with feature learning theory**. We use the classification task as an example due to the following reasons:
>
> - Most prompt fine-tuning under VLM, as demonstrated in CoOp and PromptFL [40, 16], **only conduct experiments on classification tasks**.
> - Also, in the feature learning community, the analysis mainly **focused on classification tasks** [1, 6, 7, 18]. Thus, the **analysis of image captioning and scene description** is open and can be categorized as **new work**.
> - Most federated learning on VLM research [15, 37, 27] **only conduct experiments on classification tasks**. Exploring federated learning for tasks such as image captioning and scene description can also be addressed in separate work.
>
> From these aspects, we **use a classification task to demonstrate the application of our theoretical framework** in this paper.
>
> > Question 4:  In Fig. 2.b, the best accuracy is achieved for 20 users, and the lowest accuracy is achieved for 5 users. Could you please elaborate this result? Could you please provide additional analyses for a larger number of users as well?
>
> The differences in accuracy, where 20 users achieve the highest accuracy and 5 users the lowest, are influenced by our **no-repeat data partitioning** strategy under a **few-shot learning scenario**.
> - Fewer users: Due to few-shot learning setting, each user receives a fixed number of samples. So **more users lead to more data**, facilitating better model training and higher accuracy.
> - More users: Due to the no-repeat partition strategy, the fixed total dataset size results in **less data per user**, decrease the learning efficiency and accuracy.
>
>
> Further experiments with 100 users is shown in **Figure R3** in the PDF. From this result, we observe that the performance of 100 users is lower than 50 users which also confirmed this phenomenon.

---

> ### Author Response · Authors · 2024-08-14
>
> Dear Reviewer XM5m and AC,
>
> Hope this message finds you well. As the discussion deadline approaches, we sincerely hope to receive a response to our rebuttal. We believe we have properly addressed the concerns by Reviewer XM5m as our reply to these similar concerns have been recoginized by other reviewers.
>
> We observed that the concerns by Reviewer XM5m regarding privacy issues in federated prompt learning and experiments involving a larger number of users are similar to those raised by Reviewer kpWj. They have acknowledged our response and maintained their score at 6. In our response, we explained that prompts are small vectors (16k parameters) without explcit semantic meaning passed through text encoder, which **actually helps protect privacy** compared to standard federated learning setting. Besides, we illustrated the phenomenon mentioned in Question 4, where "20 users achieve the highest accuracy," by discussing **the partition strategy used for few-shot learning**. We also included experiments with a larger number of users, as shown in **Figure R3** of the PDF.
>
> The other concern by Reviewer XM5m about generalization to different tasks was also addressed for Reviewer BdA2, who **appreciated our response and increased their score from 4 to 5**. We highlighted that the classification tasks exemplify our theoretical framework and that previous work considers only classification tasks as well. Therefore, **extending to different tasks is beyond the scope of this paper**. Additionally, we demonstrated generalization to different structures through further experiments, as shown in **Figure R2** of the PDF.
>
> Thank you again for comments. We sincerely hope you can reconsider the score for our work.
>
> Best regards,
>
> Submission 6698 Authors

---

### Official Review · Reviewer_bKME · 2024-07-13

**Soundness:** 2
**Presentation:** 2
**Contribution:** 4
**Rating:** 5
**Confidence:** 3

**Summary:**

The proposed methodology offers a novel take on analyzing federated learning using prompt learning (for foundation models) via feature learning theory. At its core, the idea is to identify and monitor task-relevant and task-irrelevant features, and leveraging inspiration from portfolio optimization which says to combine independent assets to maintain income while decreasing risk of investment, construct a 2 part prompt. One part which is global and another which is local, to help with personalization. By mixing these prompts optimally, the proposed PromptFolio method highlights an improvement in overall performance of the FL system.

**Strengths:**

- A key strength of the proposed method is its novel take on explaining prompt-based federated learning, using feature learning theory.
- Clarity in idea and its link to existing work.

**Weaknesses:**

Abstract/Conceptual Questions:
- Could the authors comment about the few-shot learning aspect of foundation models (+prompt learning) and how the optimal mixing coefficient would be affected in that scenario?
- An adjacent sub-field that contains foundation models and federated learning is the study of their impact in learning new classes using prior knowledge (base vs. new classes performance). Could you comment on how this sub-field could relate to the proposed method?

Clarifications:
- Could the authors clarify the exact dimensionality of the prompt used in each of the baselines as well as PromptFolio?
- Could the authors mention if there are previous works that have proposed a 2-level prompt learning idea? (since Section 2.2 L 107 mentions that an exploration of such a idea and the cooperation between prompts is sparse)
- In Section 4. L. 224, the authors mention that "task-relevant coefficients can be directly added". Could you clarify this statement?
- Could the authors highlight the data setting used to generate Table 1?
- Could the authors explicitly state the train and test conditions (settings) used within the experiments?

Suggestions:
- Since the proposal of PromptFL, there are have been further advances to the idea of prompt learning within the federated setting. Could the authors comment and draw comparisons against the more recent methods?

**Questions:**

The questions have been included in the weaknesses section above.

**Limitations:**

yes, the authors have addressed limitations within their choices of the experiment setting. However, I would recommend drawing more broad comparisons as to potential missing pieces in comparison to more recent work as well as the finer points from the feature learning theory that don't fully match up to the settings in the FL system.

---

> ### Author Rebuttal · Authors · 2024-08-07
>
> Thank you for your comment. We will provide a detailed, item-by-item response to these concerns.
> > Abstract/ Conceptual Question 1: Could the authors comment about the few-shot learning aspect of foundation models (+prompt learning) and how the optimal mixing coefficient would be affected in that scenario?
>
> To address your concern, we conducted additional experiments with 16, 8, 4, 2, and 1-shot learning settings. The results are shown in **Figure R1** in the PDF. These results demonstrate that the **optimal coefficients are consistent across different number of shots**, indicating the robustness of our method in few-shot learning scenarios.
>
> > Abstract/ Conceptual Question 2: An adjacent sub-field that contains foundation models and federated learning is the study of their impact in learning new classes using prior knowledge (base vs. new classes performance). Could you comment on how this sub-field could relate to the proposed method?
>
> Thank you for your suggestion. Our method can be applied to this subfield and address the high adaptation ability. In our setting, the global prompt retains global information (common prior knowledge), while the local prompt incorporates local information (novel class information). The PromptFolio mechanism allows the **aggregation of global prompts to demonstrate good generalization ability** and helps the **local prompt quickly adapt to novel classes or unique data**.
>
> > Clarification 1: Could the authors clarify the exact dimensionality of the prompt used in each of the baselines as well as PromptFolio?
>
> Each prompt in our method has 8k parameters, consisting of 64 tokens, each with 128 dimensions. We use two prompts, totaling 16k parameters. In comparison, other baselines, including CoOp, PromptFL, and the variants of PromptFL, use prompts with 8k parameters.
>
> > Clarification 2: Could the authors mention if there are previous works that have proposed a 2-level prompt learning idea? (since Section 2.2 L 107 mentions that an exploration of such a idea and the cooperation between prompts is sparse)
>
> We are afraid the sentence in Section 2.2 L 107 may lead to some confusion. In fact, what we want to express is that **"the theoretical analysis of federated prompt learning is sparse."** The absence of such an analytical framework prevents us from illustrating how two prompts cooperate.
>
> Regarding previous works, there is a similar work, FedOTP, which used two prompts combined with optimal transport for federated learning. But this approach also lacks theoretical support. Our paper aims to fill this gap by proposing the **first analytical framework for federated prompt learning**, specifically showing how constructing PromptFolio can enhance performance.
>
> > Clarification 3:  In Section 4. L. 224, the authors mention that "task-relevant coefficients can be directly added". Could you clarify this statement?
>
> In our theoretical framework, "task-relevant coefficients can be directly added" means that **task-relevant coefficients are proportional to the calculation of learnable parameters**. This statement comes from Theorem 4.3, where we demonstrate that the test performance conforms to a Gaussian distribution. Here, the mean and variance of the Gaussian distribution are task-relevant coefficients and task-irrelevant coefficients, respectively. Since the Gaussian distribution can be additive, so can the task-relevant coefficients.
>
> For example, if we have a learnable prompt $p = \theta p_1 + (1-\theta)p_2$, and $\alpha_1$ and $\alpha_2$ are corresponding task-relevant coefficients, then the task-relevant coefficient of $p$ is  $\alpha = \theta \alpha_1 + (1-\theta) \alpha_2$.
>
> > Clarification 4 & 5: Could the authors highlight the data setting used to generate Table 1? Could the authors explicitly state the train and test conditions (settings) used within the experiments?
>
> The experimental setting for Table 1 is as follows: we used the Dirichlet distribution with a parameter of $\alpha = 1$ and 16-shot training over 100 communication rounds. The training batch size was 32, and the testing batch size was 100. The learning rate was 0.001, and we used ViT-B/16 as our vision backbone.
>
> The detailed experimental settings can be referred to in our code and will be added in the final version.
>
> > Suggestions 1: Since the proposal of PromptFL, there have been further advances to the idea of prompt learning within the federated setting. Could the authors comment and draw comparisons against the more recent methods?
>
> We did analyze several recent prompt-based federated learning methods in Section 2.1, Lines 80-89, including FedPrompt, PromptFL, pFedPrompt, and pFedPG. Additionally, we conducted experiments comparing our approach with FedTPG, PromptFL, and its variants.
>
> It is worth mentioning that compared to these methods, our paper focuses **on proposing a theoretical framework for federated prompt learning** and **uses a simple but efficient algorithm** with the idea of portfolio optimization **as an example** to demonstrate the usage of our theoretical framework.
>
> > Limitations: However, I would recommend drawing more broad comparisons as to potential missing pieces in comparison to more recent work as well as the finer points from the feature learning theory that don't fully match up to the settings in the FL system.
>
> Thank you for your suggestion. We've conducted experiments with more users, different shot numbers, and different backbones, and compared our results with FedTPG. The experimental results can be referred to in the PDF.

---

> ### Comment · Reviewer_bKME · 2024-08-13
>
> I thank the authors for their in-depth responses to my comments. As a follow up to their responses,
>
> - As a natural follow-up to the responses to Q1 under clarifications, have the authors observed the change in performance of baselines when switching to prompts of 16k params?
> - Could the authors provide more context for the selection of $\alpha=1$ and discuss the impact of lower $\alpha$ on the algorithm? (if possible also provide results) I am specifically interested in the lower $\alpha$ setting since it would allow for more in-depth and interesting analyses given the theoretical framework outlines in the manuscript.

---

> > ### Author Response · Authors · 2024-08-13
> >
> > Thank you for your response. We'd like to address your concerns.
> >
> > > Q1: As a natural follow-up to the responses to Q1 under clarifications, have the authors observed the change in performance of baselines when switching to prompts of 16k params?
> >
> > 1) First, we would like to emphasize that in our algorithm, although 16k parameters (8k global + 8k local) are used, only the 8k parameters corresponding to the global prompt are shared across clients in federated learning. Thus, our comunication cost is exact the same as baselines with 8k parameters. Using baselines with 16k parameters will double the communication cost, resulting in an unfair comparison.
> >
> > 2) To further alleivate your concerns, we have added experiments by increasing the parameters of baselines from 8k to 16k. From the following table, we can obsesrve that: 1) increasing the parameters will not always increase the performance; 2) PromptFolio still has the advantage even under this unfair comparsion benefit from the prompt portfolio mechanism.
> >
> > |                      | Cifar100             | DomainNet            | Caltech10            | OxfordPets           | DTD                  |
> > | -------------------- | -------------------- | -------------------- | -------------------- | -------------------- | -------------------- |
> > | CoOp (8k param)      | 76.88 $\pm$ 0.07     | 91.83 $\pm$ 0.13     | 97.10 $\pm$ 0.20     | 87.85 $\pm$ 0.32     | 56.39 $\pm$ 0.48     |
> > | CoOp (16k param)     | 77.74 $\pm$ 0.21     | 91.92 $\pm$ 0.18     | 96.41 $\pm$ 0.43     | 87.68 $\pm$ 0.44     | 58.33 $\pm$ 0.41     |
> > | PromptFL (8k param)  | 78.16 $\pm$ 0.16     | 92.72 $\pm$ 0.16     | 95.51 $\pm$ 2.62     | 88.91 $\pm$ 0.72     | 70.99 $\pm$ 0.32     |
> > | PromptFL (16k param) | 78.53 $\pm$ 0.33     | 92.88 $\pm$ 0.29     | 96.23 $\pm$ 1.41     | 91.81 $\pm$ 0.64     | **71.73 $\pm$ 0.31** |
> > | PromptFolio (8k + 8k param) | **80.17 $\pm$ 0.05** | **93.04 $\pm$ 0.09** | **97.24 $\pm$ 0.11** | **92.17 $\pm$ 0.32** | 71.32 $\pm$ 0.49     |
> >
> >
> > > Q2: Could the authors provide more context for the selection of $\alpha = 1$ and discuss the impact of lower $\alpha$ on the algorithm? (if possible also provide results) I am specifically interested in the lower $\alpha$ setting since it would allow for more in-depth and interesting analyses given the theoretical framework outlines in the manuscript.
> >
> >
> > We conducted experiments with multiple different $\alpha$ values instead of using only $\alpha = 1$. For instance, in Table 2, we used $\alpha = 0.3$ to compare our result with baseline methods. In Figure 2(a), we conducted parameter analysis for $\alpha$ by implementing PromptFolio with different $\alpha$ values (0.01, 0.1, 1, 10). Here, we observe that a lower $\alpha$ indicates a more non-i.i.d. data distribution, requiring a larger mixing coefficient $\theta$ to retain more local information.
> >
> > Our theoretical analysis also supports this result. In our framework, $\Chi_k$ measures the heterogeneity across clients. A lower $\alpha$ leads to more data heterogeneity, meaning $\Chi_k$ will be lower as well. As described in Theorem 5.2 on Line 279, a lower $\Chi_k$ (lower $\alpha$) corresponds to a higher $\theta^\star$, also aligning with the experimental results in Figure 2. Further analysis on how different levels of data heterogeneity affect learning dynamics can be found in Theorem D.2.

---

> > > ### Author Response · Authors · 2024-08-14
> > >
> > > Dear Reviewer bKME,
> > >
> > > We thanks for your comments again. As the discussion process is coming to a close, please let us know if you have any further questions.
> > >
> > > Best regards,
> > >
> > > Submission 6698 Authors

---

### Official Review · Reviewer_BdA2 · 2024-07-18

**Soundness:** 3
**Presentation:** 3
**Contribution:** 3
**Rating:** 5
**Confidence:** 4

**Summary:**

This paper tackles the interesting problem of analyzing federated learning (FL) from vision-language foundation models like CLIP. The authors develop a theoretical framework based on feature learning theory to understand how prompt-based FL works. They introduce a new algorithm called PromptFolio that mixes global and local prompts, drawing an analogy to portfolio optimization in finance. The theoretical analysis shows how this approach can balance generalization and personalization in FL. They back up their claims with experiments on several datasets.

**Strengths:**

- The feature learning approach they develop seems pretty solid.
- The connection between prompt mixing and financial portfolio theory is novel.
- The ablation studies on data heterogeneity and client numbers are particularly informative.

**Weaknesses:**

- This reviewer is confused by the feature learning theory (I am not familiar with it), which is introduced in a relatively short section. Is it (looks like) a common sense in deep learning or a kind of new theory proposed recently? It would be helpful to have a more thorough introduction to the theory.
- Using a single activation function for the text encoder is a pretty big simplification from how CLIP and other large language models actually work.
- All the experiments focus on classification tasks. Any thoughts on how PromptFolio might perform on other vision-language tasks like image captioning or visual QA in a federated setting?
- Why is the proposed method on OxfordPets dataset not as good as coop? While it seems to be dominating on other datasets.

**Questions:**

see weaknesses.

**Limitations:**

yes

---

> ### Author Rebuttal · Authors · 2024-08-07
>
> We sincerely thank you for your comments and questions. We appreciate that you found our feature learning approach "solid," the connection between prompt mixing and portfolio optimization "novel," and our ablation studies "informative."
>
> > Weakness 1: This reviewer is confused by the feature learning theory (I am not familiar with it), which is introduced in a relatively short section. Is it (looks like) a common sense in deep learning or a kind of new theory proposed recently? It would be helpful to have a more thorough introduction to the theory.
>
> Feature Learning theory is a **new theoretical framework** proposed recently. The seminal work [1] (here we reuse the citations in the paper due to word limit) proposed feature learning theory to understand ensemble, knowledge distillation, and self-distillation in deep learning. They show that **when data has a structure containing features, the network's dynamics can be tracked during gradient training, which can further be used to characterize generalization**. Later, this theory was further systematized and standardized by [6], forming a fundamental theoretical framework to explain benign overfitting. Since then, feature learning theory has been widely used to understand algorithms [7, 18] and techniques [10, 17] in deep learning, forming a comprehensive theoretical system.
>
> To provide more details on the general feature learning process:
> 1. **Defining the feature space**: As outlined in Section 4, lines 145-155, features are categorized as task-relevant and task-irrelevant. For example, in an image of a cat, features representing the cat itself are task-relevant, while background features are task-irrelevant.
> 2. **Parameter representation**: As described in Lemma 4.1, learnable parameters can be decomposed into the feature space. In our framework, the learnable prompts are expressed as a linear combination of task-relevant and task-irrelevant features.
> 3. **Learning dynamics**: Theorem 4.2 examines the learning dynamics of coefficients of feature learning, where coefficients are defined by the weight decomposition into the feature space, providing valuable insights into the learning process. When task-relevant features dominate, the network learns the target effectively and demonstrates good performance.
> 4. **Generalization bound**: By leveraging the learning dynamics of the coefficients, we can demonstrate the generalization bound or test performance post-training. In our work, we connect the performance of prompt fine-tuning with the ratio between task-relevant coefficients and task-irrelevant coefficients.
>
> > Weakness 2: Using a single activation function for the text encoder is a pretty big simplification from how CLIP and other large language models actually work.
>
> Actually, we also admit this simplification as a limitation in the Conclusion section. It is worth mentioning that the theoretical analysis of CLIP is sparse and very challenging. Very recently, an initial theoretical work [8] analyzing CLIP published at ICLR 2024 even assumes a **linear layer** as the text encoder. As a result, our approach incorporating nonlinear activation functions is more **suitable for CLIP analysis** and is **non-trivial**.
>
> Also, in feature learning theory [1,6,18], the simplification of the network is necessary for theoretical tractability. Therefore, we also adopt a network simplification here and only use it as a theoretical assumption.
>
> Despite this, our **experimental results align well with our theoretical predictions**, suggesting the practicality of our framework. Future work will aim to extend these theoretical tools to more complex and practical network architectures.
>
> > Weakness 3: All the experiments focus on classification tasks. Any thoughts on how PromptFolio might perform on other vision-language tasks like image captioning or visual QA in a federated setting?
>
> Thank you for your suggestion. Firstly, it is worth mentioning that our paper mainly focuses on proposing the **first theoretical framework for federated prompt learning with feature learning theory**. We use the classification task as an example due to the following reasons:
>
> - Most **prompt fine-tuning under VLM**, as demonstrated in CoOp and PromptFL [40, 16], only conduct experiments on classification tasks.
> - Also, in the **feature learning community**, the analysis mainly focused on classification tasks [1, 6, 7, 18]. Thus, the analysis of image captioning and VQA is open and can be categorized as new work.
> - Most **federated learning on VLM** research [15, 37, 27] only conduct experiments on classification tasks. Exploring federated learning for tasks such as image captioning and VQA can also be addressed in separate work.
>
> From these aspects, we **use a classification task to demonstrate the application of our theoretical framework** in this paper.
>
> **Any thoughts:** We found that the idea of PromptFolio could potentially be applied to federated image captioning and VQA tasks. By combining global and local parameters, this approach is expected to achieve a **good balance between personalization and generalization**.
>
> > Weakness 4: Why is the proposed method on OxfordPets dataset not as good as coop? While it seems to be dominating on other datasets.
>
> The discrepancy in performance on the OxfordPets dataset arises mainly due to:
> 1. In our experiment, **we use a uniform mixing coefficient** $\theta = 0.2$ for all datasets. However, the optimal coefficient may vary across datasets.
> 2. The inherent high variance observed in CoOp results. The **high variance of the CoOp** results (1.30) may lead to statistical errors during comparison.
>
> Thus, we fine-tune the coefficient of PromptFolio and rerun the experiment of CoOp, as detailed in **Table R1** in the PDF. As a result, our PromptFolio currently outperforms CoOp and demonstrates consistency across datasets.

---

> > ### Comment · Reviewer_BdA2 · 2024-08-14
> >
> > Thanks for the rebuttal.
> >
> > Some of my concerns have been addressed.
> >
> > I would raise my score.

---

> > > ### Author Response · Authors · 2024-08-14
> > >
> > > Thank you for reconsidering our work and raising your score. We greatly appreciate your constructive feedback and support.

---

### Author Rebuttal · Authors · 2024-08-07

We sincerely thank all the reviewers for their time and effort in evaluating our work and providing constructive comments. We appreciate that the reviewers consider our theoretical framework for federated prompt learning to be "pretty solid" (BdA2), "thorough and clear" (YEHg), and "novel" (bKME). The relation between prompt mixing and portfolio optimization is viewed as "novel" (BdA2) and a "nice idea" (Xm5m). The ablation study towards data heterogeneity and client numbers is particularly informative (BdA2). We are further glad that the reviewers agree our work is interesting and useful for the community (kpWj).

In the following, we will try to address some common concerns/questions of the reviewers and provide a detailed item-by-item response to these concerns.

> Q1: The novelty of this paper.

Our paper mainly focuses on developing **a theoretical framework for federated prompt learning**. This theoretical framework introduces feature learning theory, decomposing the learnable prompts into task-relevant and task-irrelevant features. By a thorough analysis of the dynamics of feature coefficients, we demonstrate the **convergence and generalization of federated prompt learning in Theorem C.5 and Theorem D.5** by evaluating the ratio between the coefficients of task-relevant and task-irrelevant features.

Under this theoretical framework, we further construct the portfolio optimization method for federated prompt learning, i.e., PromptFolio, and **derive an analytical solution within this theoretical framework**. PromptFolio seeks to balance the global consensus and local personalization of federated prompt learning. This simple but efficient method can even outperform many SOTA federated prompt learning methods, such as PromptFL, FedTPG, PromptFL+FedProx, PromptFL+FedAMP, etc.

> Q2: How do prompt learning assure privacy protection?

Actually, prompt learning can protect privacy in federated learning. It is worth mentioning that the prompt here is a sequence of dense vectors sent to the text encoder with only 8K parameters. This small number of parameters make it **difficult to recover the original data information**. In our proposed method, the prompts that encapsulate information in a highly abstract form are sent to the server while the **sensitive data are kept on the local client**. Thus, there are no privacy concerns in prompt learning.

> Q3: Why you use classification task as an example?

Classification, as the most popular tasks, is widely used for a machine learning theoretical paper. In addition, most existing works in the area we consider only test on classification tasks.

- Most **prompt fine-tuning under VLM**, as demonstrated in CoOp and PromptFL [40, 16], only conduct experiments on classification tasks.
- Also, in the **feature learning community**, the analysis mainly focused on classification tasks [1, 6, 7, 18].
- Most **federated learning on VLM** research [15, 37, 27] only conduct experiments on classification tasks.
---
**Illustration of the attached PDF file:**

The attached PDF file includes one table and three figures.

Table R1 compares our PromptFolio with other state-of-the-art algorithms, including a comparison with FedTPG as requested by Reviewer YEHg. This table shows that PromptFolio performs well across various datasets. The experiments were conducted using a Dirichlet distribution with $\alpha = 0.3$ over 100 communication rounds. We used the ViT-B/16 backbone, with a training batch size of 32 and a test batch size of 100.

Figure R1 presents experimental results for different numbers of shots, following Reviewer bKME's suggestion. We conducted experiments with 16, 8, 4, 2, and 1 shots, demonstrating that the optimal coefficients are consistent across different shot numbers, indicating the robustness of our method in few-shot learning scenarios.

Figure R2 shows experimental results for different backbones, as per Reviewer XM5m's recommendation. We tested different backbones, including ViT-B/16 and ResNet-50. The results confirm our theoretical analysis, validating the application of our theoretical framework.

Figure R3 includes experiments on different users. We've added results for 100 users according to Reviewer XM5m and kpWj. The experiment demonstrates the consistency of our theoretical framework when applied to a larger number of users.

---

### Author Response · Authors · 2024-08-12
**Reviewer Feedback Request on Rebuttal**

Dear Reviewers and AC,

Thank you for your valuable comments. We have made every effort to address the concerns raised by the reviewers.

We notice that none of the reviewers have provided follow-up comments on our responses. As the discussion stage deadline approaches, please let us know if you have any further questions.


Thank you and best regards,
Submission 6698 Authors

---

### Decision · Program_Chairs · 2024-09-25

**Decision:**

Accept (poster)

**Comment:**

The paper proposes a theoretic framework for understanding prompt-based FL based on feature learning theory [1].
They then draw an analogy to portfolio optimization from the finance field and prove that a mixture of global and local prompts can balance between generalization and personalization.

The reviewers generally agreed that the connection to feature learning theory and the resulting theoretic method was novel and insightful. One of the key challenges was appropriately explaining feature learning theory for new readers and properly framing the main contributions. In the revision, please include better introduction, background, and intuition for the feature learning theory to aid new readers.

There were some additional minor concerns that this was only evaluated on the classification task but this seems reasonable given that the paper aims towards theoretic contributions primarily. Given the confusion, I would recommend revision of the paper to more carefully emphasize the theoretic focus of the paper and deemphasize the algorithmic novelty.

Overall, the paper provides theoretic insight into the federated prompt learning. This theoretic insight is used to develop a simple but effective strategy for combining local and global prompts. Because this paper adds some theoretic grounding for federated prompt learning, the paper provides a valuable contribution to the NeurIPS community.

[1] Allen-Zhu, Z., & Li, Y. Towards Understanding Ensemble, Knowledge Distillation and Self-Distillation in Deep Learning. In The Eleventh International Conference on Learning Representations.